# Immune-profiling of ZIKV-infected patients identifies a distinct function of plasmacytoid dendritic cells for immune cross-regulation

Xiaoming Sun[1,4], Stephane Hua [1,4], Ce Gao [1], Jane E. Blackmer[1], Zhengyu Ouyang[1], Kevin Ard[2], Andrea Ciaranello[2], Sigal Yawetz[3], Paul E. Sax[3], Eric S. Rosenberg[2], Mathias Lichterfeld[1,3] & Xu G. Yu[1,3 ✉]

Zika virus (ZIKV) is a mosquito-borne pathogen with increasing public health significance. To characterize immune responses to ZIKV, here we examine transcriptional signatures of CD4 T, CD8 T, B, and NK cells, monocytes, myeloid dendritic cells (mDCs), and plasmacytoid dendritic cells (pDCs) from three individuals with ZIKV infection. While gene expression patterns from most cell subsets display signs of impaired antiviral immune activity, pDCs from infected host have distinct transcriptional response associated with activation of innate immune recognition and type I interferon signaling pathways, but downregulation of key host factors known to support ZIKV replication steps; meanwhile, pDCs exhibit a unique expression pattern of gene modules that are correlated with alternative cell populations, suggesting collaborative interactions between pDCs and other immune cells, particularly B cells. Together, these results point towards a discrete but integrative function of pDCs in the human immune responses to ZIKV infection.

[1] Ragon Institute of MGH, MIT, and Harvard, Massachusetts General Hospital, Boston, MA, USA. [2] Infectious Disease Division, Massachusetts General Hospital, Boston, MA, USA. [3] Infectious Disease Division, Brigham and Women's Hospital, Boston, MA, USA. [4]These authors contributed equally: Xiaoming Sun, Stephane Hua. ✉email: xyu@mgh.harvard.edu

Zika virus (ZIKV), an enveloped, positive-stranded RNA virus belonging to the *Flaviviridae* family, was first isolated in the Zika Forest of Uganda in 1947 (ref. [1]). Similar to most flaviviruses, ZIKV is predominantly spread by *Aedes spp.* mosquitoes. However, sexual transmission and mother-to-child transmission have also been reported, specifically in the recent ZIKV outbreak in the Americas[2,3]. Clinical symptoms of ZIKV infection are typically nonspecific, although neurological birth malformations such as microcephaly have recently emerged as serious complications of fetal or neonatal ZIKV infection[4–6]. There is currently no definitive therapy available to treat ZIKV infection, but symptoms resolve spontaneously in the majority of cases, typically without residual symptoms. Yet, some reports suggest persistence of virally infected cells for prolonged periods of time[7–9], implying the existence of viral cell and tissue reservoirs that resist host immune clearance. Several vaccine candidates for ZIKV are currently under development[10–12], although the precise correlates of effective ZIKV immune protection remain incompletely understood.

Akin to other viral pathogens, ZIKV seems to have developed a finely tuned preference for specific target cells that offer immune microenvironments most conducive to viral replication steps. Preferred target cells for ZIKV include placental cells[13], neuro-progenitor cells[14], and circulating myeloid cells[15,16], all of which express high levels of the ZIKV entry receptor AXL. The choice of dendritic cells (DCs) and monocytes as preferred target cells for ZIKV seems paradoxical, as these cells harbor a complex network of innate immune sensors that can recognize viral nucleic acids and activate potent type I interferon (IFN-I)-dependent cell-intrinsic antiviral immune defense programs. These IFN-I-dependent immune responses arguably represent the most critical immune component for restricting ZIKV replication, as evidenced by the high susceptibility of IFN-α receptor-deficient research animals to ZIKV[17,18], and by the increased ZIKV-associated cytopathic effects typically observed in cells lacking a cell-intrinsic antiviral IFN-I response[19,20]. Nevertheless, ZIKV seems to have developed multiple strategies to counterbalance host immune mechanisms, at least in some cell types. For instance, in some viral target cells, ZIKV can inhibit phosphorylation and promote proteasomal degradation of human STAT2, a key downstream regulator of IFN-I responses[19]. Moreover, there is evidence that ZIKV can reprogram the transcriptional signatures of myeloid DCs (mDCs) in favor of the virus by altering global gene expression patterns in a way that facilitates viral replication and reduces cell-intrinsic antiviral immune defenses[16]. Therefore, ZIKV may have the ability to transform otherwise hostile mDCs into territories supportive of ZIKV replication.

Despite the substantial progress in understanding ZIKV-specific immune responses in individual target cells, very little is known about concerted ZIKV-specific antiviral immune responses that are mediated through systemic and collaborative interactions between individual cell types. Such responses are particularly difficult to analyze in infected humans due to challenges in identifying individuals with active ZIKV infection and limitations in the numbers of cells that are available from such individuals. In the present work, we have conducted a parallel, unbiased transcriptional profiling analysis of mDCs, plasmacytoid DCs (pDCs), natural killer (NK) cells, B cells, CD4 T cells, and CD8 T cells from three individuals who were diagnosed with ZIKV infection during the 2016–2017 ZIKV disease outbreak in the Caribbean. Our data show that in contrast to mDCs, pDCs are poorly susceptible to replicative ZIKV infection and express a distinct immune signature that differs from alternative cells by a strong upregulation of IFN-I-dependent genes combined with a downregulation of critical ZIKV-dependency genes (ZDGs). In addition, gene expression signatures in pDCs demonstrated discrete co-linearities with transcriptional profiles of other cell types, suggesting collaborative immunological circuits between pDCs and other cells that are initiated in response to ZIKV infection. Together, these data point to a central role of pDCs in immune defense during acute ZIKV infection in humans.

## Results

**Transcriptional changes in circulating immune cells.** Our previous studies demonstrated that in individuals with naturally acquired ZIKV infection, *ZIKV* RNA was detectable in mDCs, but not in pDCs, suggesting that cellular susceptibility and cell-intrinsic immune responses to ZIKV may differ among individual immune cell subsets[16]. To gain systemic insight into the immune response caused by ZIKV infection in humans, we conducted RNA sequencing (RNA-Seq)-based transcriptional profiling experiments to characterize gene expression changes in seven immune cell populations (CD4 T cells, CD8 T cells, B cells, NK cells, monocytes, mDCs, and pDCs) from the peripheral blood of three study individuals with acute ZIKV infection; cells from three gender- and age-matched healthy individuals were treated identically and were used as reference samples. Clinical characteristics of these study individuals were described in our previous study[16] and Supplementary Table 1. We observed that on a global transcriptional level, gene expression signatures differed profoundly among the individual cell populations. Specifically, NK and CD8 T cells showed relatively minor transcriptional differences between ZIKV-infected patients and controls, with less than 300 transcripts meeting our criteria for differential expression (false discovery rate (FDR)-adjusted $p < 0.05$, fold change in gene expression intensity > 1.5) (Fig. 1). However, in the remaining cell types, ZIKV infection was associated with a markedly altered transcriptional profile, with numbers of differentially expressed genes (DEGs) ranging from 600 in monocytes to more than 3000 in B cells (Fig. 1a, b and Supplementary Data 1). Notably, although DEGs were mainly downregulated in most cell subsets from ZIKV-infected individuals, pDCs showed more balanced transcriptional alterations, with similar numbers of upregulated and downregulated DEGs (Fig. 1b, d).

A computational canonical pathway analysis of DEGs predicted that key functional entities, including those involved in Fc receptor-mediated phagocytosis, Toll-like receptor (TLR) signaling, and cytokine and mammalian target of rapamycin (mTOR) signaling, were mostly inhibited in B cells, monocytes, and mDCs from ZIKV-infected patients (Fig. 2a and Supplementary Data 2). Consistent with these results, no functions were predicted to be induced in B cells, CD4 T cells, monocytes, and mDCs based on annotation enrichment analysis of their DEGs using Ingenuity Pathway Analysis (IPA) (Fig. 2b). In striking contrast, we noticed that critical functional activities of pDCs, such as IFN signaling, TLR signaling, innate immune recognition, and interleukin (IL)-8 secretion and IL-8 signaling were predicted to be activated in pDCs from ZIKV-infected patients. Moreover, this computational analysis also inferred an enrichment in pDCs with transcripts involved in endocytosis, phagocytosis, activation of antigen-presenting cells and recruitment of blood cells (Fig. 2a, b and Supplementary Data 2). Notably, upstream regulators predicted to govern transcriptional changes in pDCs during ZIKV infection included members of the tumor necrosis factor family, colony-stimulating factors (CSF1–3) and IL-4; for all of these cytokines, an involvement in transcriptional regulation of alternative cell populations was markedly less obvious (Fig. 2c and Supplementary Data 3). Together, these results suggest a distinct response of pDCs to ZIKV infection in humans.

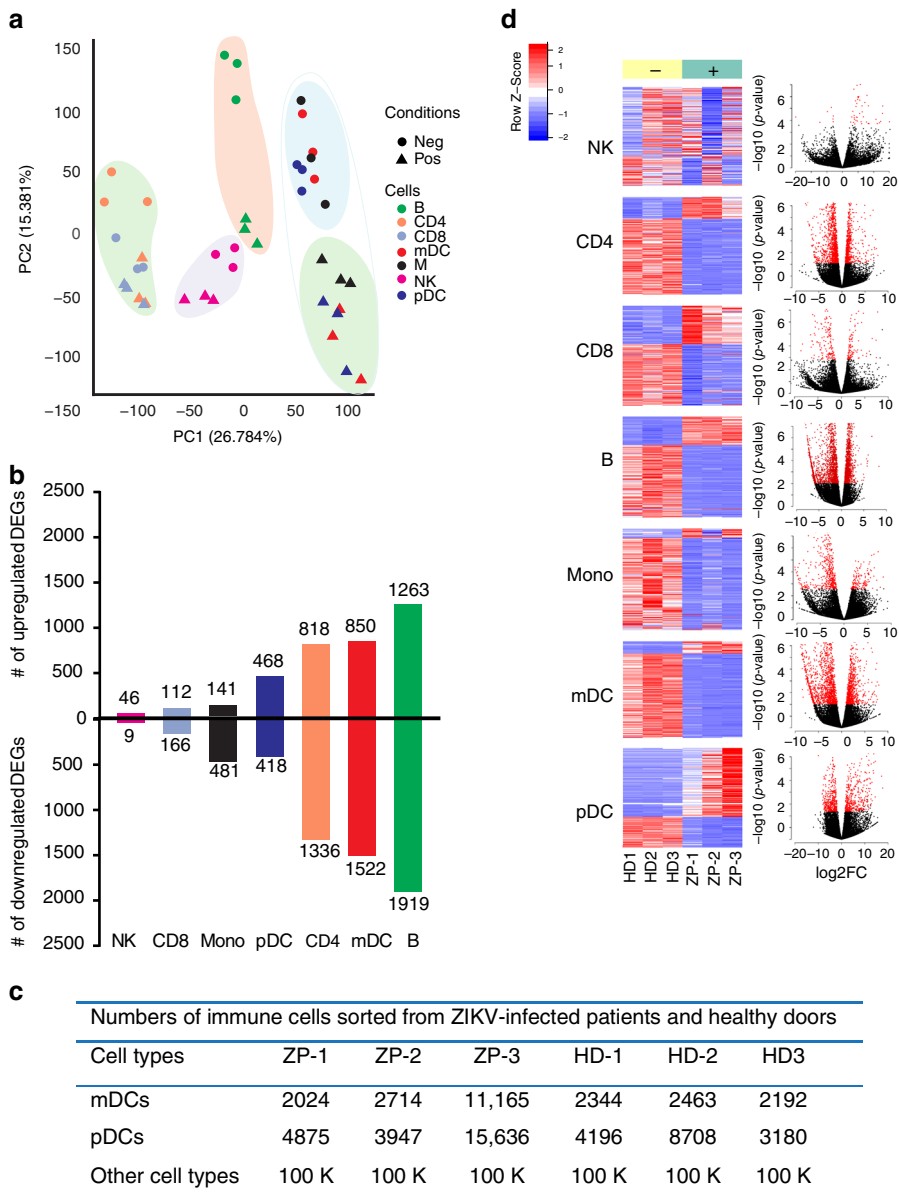

**Fig. 1 Transcriptional signatures of immune cell subpopulations during naturally acquired ZIKV infection. a** Principal component analysis of global transcriptional signatures from indicated cell populations derived from ZIKV-infected (positive) and ZIKV-noninfected (negative) control individuals. **b** Bar diagram indicating the total number of upregulated and downregulated genes in indicated cell populations from ZIKV-infected patients, relative to noninfected individuals (FC = 1.5, FDR < 0.05). **c** Numbers of cells sorted for indicated cell populations from ZIKV-infected patients and noninfected individuals. **d** Heatmaps demonstrating transcriptional patterns of top100 differentially expressed genes (DEGs) between ZIKV-infected individuals and noninfected individuals in indicated immune cell subsets (left panel). Volcano plots demonstrate fold change (FC) in expression intensity of DEGs, plotted against corresponding $p$-values (right panel).

**Expression of ZIKV-dependency genes and IFN-stimulated genes.** It is well-recognized that effective ZIKV replication depends on a substantial number of host proteins that enable and support individual steps of the viral life cycle in susceptible cells. ZIKV may alter and reprogram the expression profile of these viral dependency genes in a way that renders immune cells more susceptible and conducive to viral replication[21]. To explore how ZIKV can influence the expression profile of viral dependency genes in different immune cell populations, we selectively identified previously described ZDGs[21] within all DEGs in each of the analyzed cell types. We observed that 165, 30, 129, and 42 viral dependency genes were differentially expressed in B cells, monocytes, mDCs, and pDCs, respectively (Fig. 3a and Supplementary Data 4) compared with uninfected controls; among these

transcripts were five common genes that were differentially expressed in all four subsets: *ARAF*, *CS*, *LAPTM5*, *LIMD2*, and *ZYX*. Of note, the expression pattern of viral dependency genes was markedly different among the studied cell populations; while B cells, monocytes, and mDCs shared similar directional changes in the expression profile of most differentially expressed viral dependency genes, expression signatures in pDCs were different for a considerable number of these transcripts (Fig. 3b). Moreover, a gene set enrichment analysis (GSEA) demonstrated that DEGs in pDCs were negatively enriched for ZDGs (normalized enrichment score (NES) of −0.82, FDR-adjusted $q$-value = 0.024), in contrast to mDCs and monocytes in which no significant enrichment or de-enrichment of ZDGs was observed (Supplementary Data 5). Notably, we observed that *AXL* and

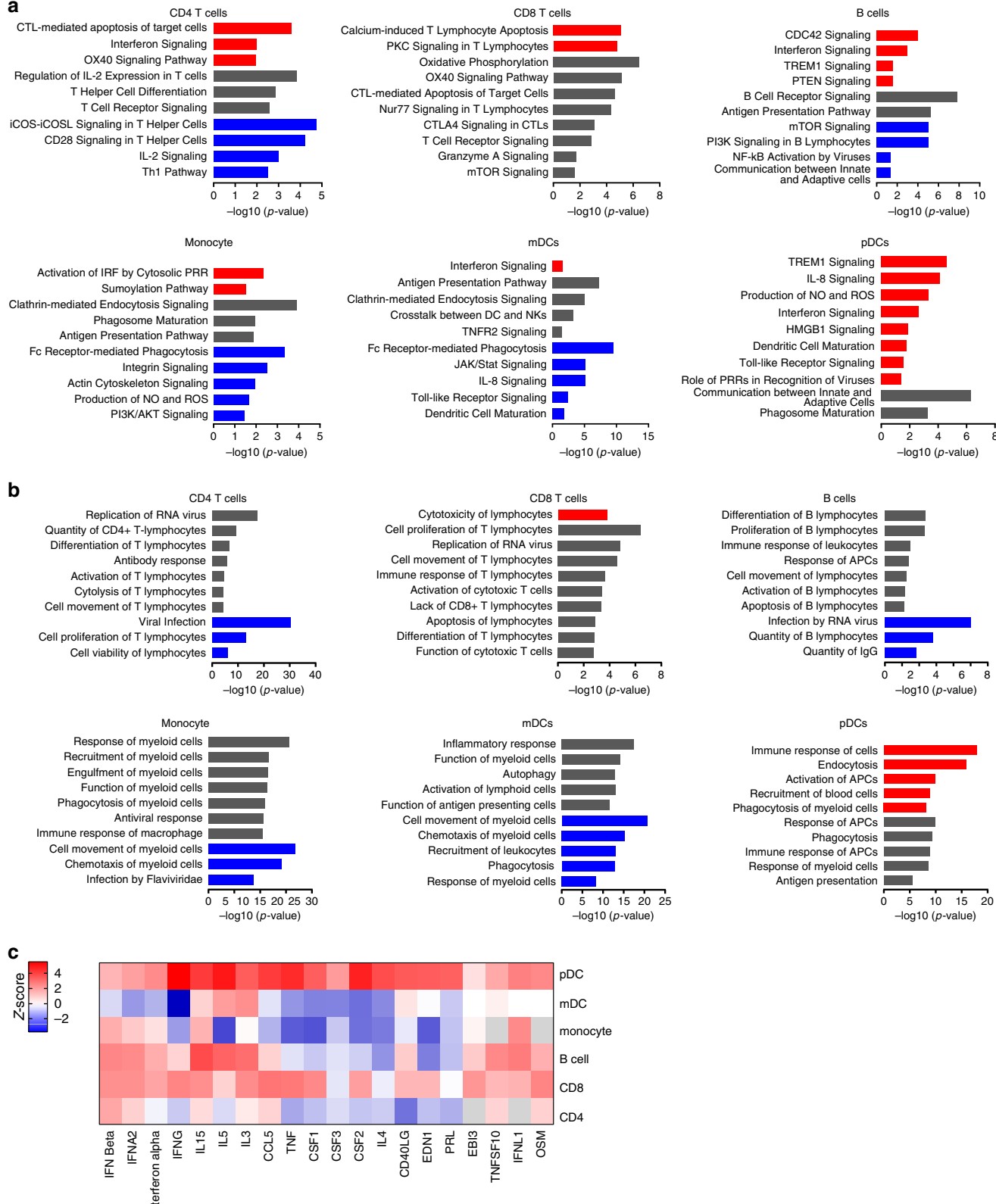

**Fig. 2 Activation of antiviral immune defense pathways in pDCs from patients with ZIKV infection. a, b** "Canonical pathways" (**a**) and "Diseases and Functions" (**b**) inferred by Ingenuity pathway analysis (IPA) from DEGs between ZIKV-infected individuals and noninfected individuals in indicated immune cell subsets. Red and blue denote functional pathways predicted to be up or downregulated, respectively; gray indicates indeterminate directional changes for the respective functional entity. **c** Predicted upstream regulators for genes differentially expressed between pDCs from ZIKV-infected individuals and noninfected individuals. Color coding reflects z-score, indicating activation or inhibition of the upstream regulators in ZIKV-infected patients compared to noninfected individuals.

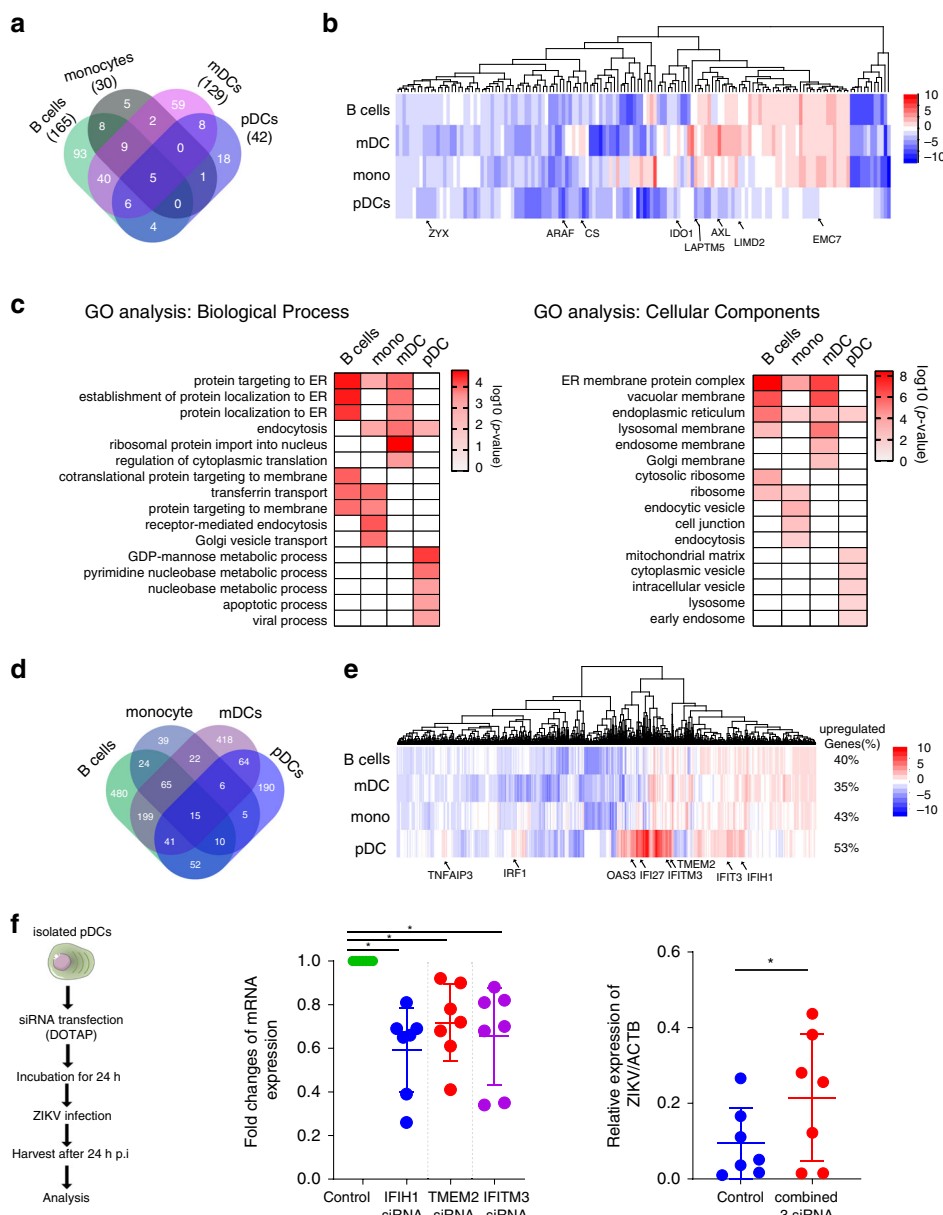

**Fig. 3 Differential expression of ZIKV-dependency genes and interferon-stimulated genes in pDCs from ZIKV-infected patients. a** Venn Diagram indicating the number of previously described ZIKV-dependency genes that meet criteria for differential gene expression between ZIKV-infected patients and noninfected individuals in indicated cell populations. **b** Heatmap reflecting expression patterns of selected ZIKV-dependency genes in indicated cell populations. **c** Computational Gene Ontology (GO) analysis of differentially expressed ZIKV-dependency genes in indicated cell populations, as determined using WebGestalt[66]. **d** Venn diagram reflecting the frequency of Interferon-stimulated genes (ISGs) meeting criteria for differential expression between ZIKV patients and controls in indicated cell populations. **e** Heatmap indicating expression patterns of ISGs meeting criteria for differential expression between ZIKV-infected patients and noninfected individuals in indicated cell populations. Proportion of upregulated/downregulated ISGs in each cell compartment is indicated. **f** Left panel: Graphical representation of in vitro ZIKV infection experiments following siRNA-mediated gene silencing in isolated pDCs. (middle panel): relative mRNA expression of *IFIH1*, *TMEM2*, and *IFITM3* mRNA in pDCs at 24 h after transfection with indicated siRNAs. Right panel: Expression of *ZIKV* RNA relative to β-actin mRNA in pDCs transfected with a cocktail of gene-specific siRNAs (targeting *IFIH1*, *TMEM2*, and *IFITM3*) or control siRNA (*n* = 7 biologically independent samples). Horizontal lines reflect mean ± SD. Statistical significance between the different subsets was tested using a Friedman test with post-hoc Dunn's test. NS, not significant; *$p < 0.05$.

*IDO1*, considered as essential host factors for ZIKV entry and replication[16,21,22], were significantly upregulated in mDCs from ZIKV patients, whereas the ER membrane protein complex subunit 7, known to support ZIKV replication through an as of yet unidentified mechanism, exhibited significantly elevated expression in monocytes and B cells from ZIKV-infected patients. In contrast, these three transcripts displayed unchanged expression in pDCs from ZIKV-infected patients relative to those from

uninfected hosts. We subsequently entered differentially expressed ZDGs from pDCs into a Gene Ontology (GO) analysis platform to computationally identify the functional gene sets most profoundly altered by ZIKV infection. Consistent with our previous results, this analysis demonstrated a distinct pattern of predicted functionalities for differentially expressed ZDGs in pDCs, with a selectively inferred enrichment of pDC transcriptional signatures with features of mitochondrial and lysosomal

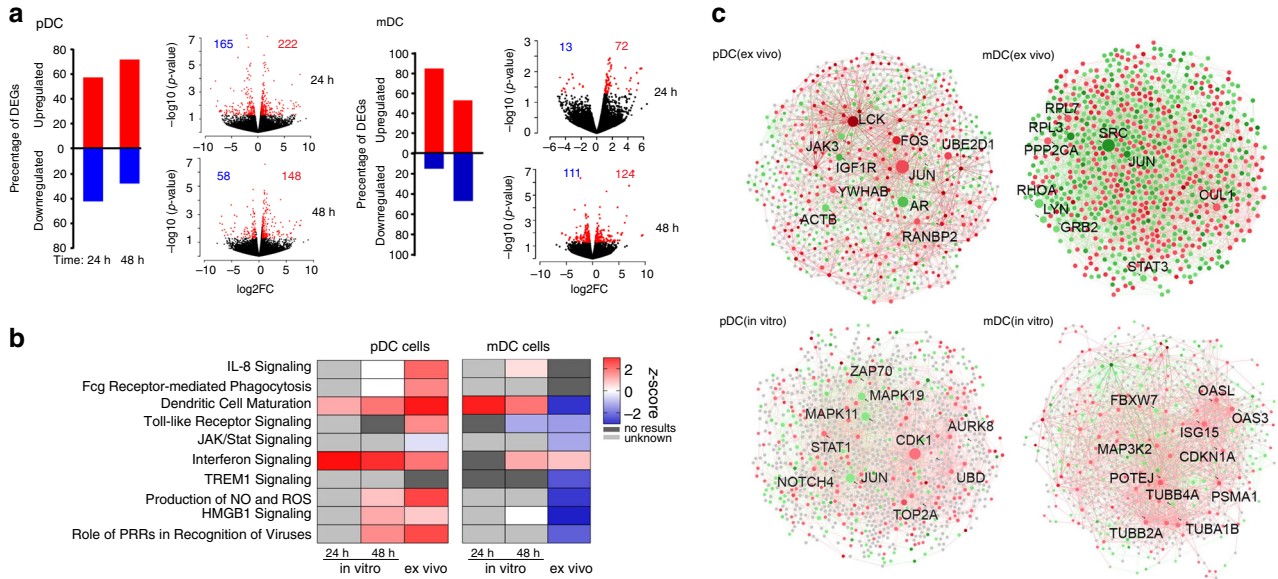

**Fig. 4 Effective restriction of ZIKV infection in pDCs. a** Bar diagrams indicating the proportion of upregulated (red) and downregulated (blue) transcripts in mDCs and pDCs at 24 h and 48 h after in vitro exposure to ZIKV. Volcano plots demonstrate the fold change in gene expression intensity of DEGs, plotted against the corresponding p-value. **b** Canonical pathway analysis (determined by IPA) of DEGs in pDCs and mDCs at indicated time points after in vitro exposure to ZIKV (in vitro), or in cells derived from ZIKV-infected individuals (ex vivo). **c** Network of differentially expressed genes in pDCs and mDCs plotted using the STING interactome database. Data from ex vivo and in vitro analysis are shown. Visualization was performed using Networkanalyst[62]. Upregulated genes are indicated in red and downregulated genes in blue. Top10 genes with highest connectivity scores are considered hub genes and indicated in the figure.

metabolic processes that may influence susceptibility to ZIKV infection (Fig. 3c).

In addition to the expression profile of viral dependency genes, cell susceptibility to ZIKV infection is also critically influenced by IFN-stimulated genes (ISGs) that may restrict ZIKV replication through a variety of mechanisms. We observed that in ZIKV-infected patients, 886, 186, 830, and 383 ISGs were differentially expressed in B cells, monocytes, mDCs, and pDCs, respectively (Fig. 3d and Supplementary Data 4) in comparison with the noninfected controls. Notably, several ISGs with described roles in restriction of ZIKV replication in different subcellular compartments, including *IFIH1*, *IFITM3*, and *TMEM2* (ref. [23]), were significantly upregulated in pDCs, in contrast to alternative cell compartments (Fig. 3e); moreover, for additional ISGs (*TNFAIP3*, *IRF1*, *OAS3*, *IFI27*, and *IFIT3*), we noted a trend for increased expression in pDCs relative to other immune cells. Resonating with these observations, a gene set enrichment analysis demonstrated a positive enrichment in pDCs with ISGs (NES of 2.46, FDR-adjusted q-value < 0.001), whereas a negative enrichment with ISGs was found in mDCs (NES of −1.59, FDR-adjusted q-value = 0.025), and, to a lesser degree, in B cells (ES: −1.38, FDR-adjusted q-value: 0.09) (Supplementary Data 5). Moreover, a computational analysis of inferred upstream regulators governing transcriptional signatures of DEGs supported a critical role of type I IFN responses in pDCs from ZIKV-infected patients, in contrast to mDCs in which type I IFN signatures appeared to be reduced (Supplementary Fig. 1b).

To further explore the functional role of ISGs for restriction of ZIKV replication, we analyzed possible antiviral effects of the three ISGs (*IFIH1*, *TMEM2*, and *IFITM3*), which were significantly upregulated in pDCs compared with other immune cell compartments. Selective transfection of pDCs with small interferingRNAs (siRNAs) individually targeting *IFIH1*, *TMEM2*, or *IFITM3* resulted in a 34%, 48%, and 36% relative reduction of mRNA expression of the target genes, respectively, but did not notably impact ZIKV replication in pDCs (Supplementary

Fig. 1d), possibly due to insufficient efficacy of siRNA-mediated gene silencing in primary pDCs. Yet, combined transfection of siRNAs directed towards all three different target ISGs (*IFIH1*, *TMEM2*, and *IFITM3*) induced a significant increase in ZIKV replication in pDCs, consistent with an important role of these ISG for restriction of ZIKV infection in pDC (Fig. 3f and Supplementary Fig. 1d). Together, these results suggest a distinct downregulation of viral dependency genes, coupled with an upregulation of ISGs in pDCs during ZIKV infection, and highlight a distinct role of pDCs in resisting and restricting ZIKV infection.

**Distinct transcriptional programs in pDCs infected in vitro.** To characterize cell-intrinsic immune responses to ZIKV in more detail, we subsequently performed transcriptional profiling experiments with isolated pDCs from healthy peripheral blood mononuclear cells (PBMCs) that were infected in vitro with ZIKV and subjected to RNA-Seq assays at indicated time points post infection. Uninfected control cells were treated identically and analyzed in parallel; there was no difference in cell viability or relative proportions of mDCs and pDCs in infected vs. non-infected PBMC (Supplementary Fig. 4b). We observed that negative-stand ZIKV RNA, which is indicative of active ZIKV replication, was selectively detected in mDC but not in pDC in these in vitro infection assays, using reverse-transcription PCR (RT-PCR) (Supplementary Fig. 2a). The number of host transcripts with significant changes in pDCs exposed to ZIKV (compared with noninfected control cells) was rather modest, with a maximum of 387 transcripts showing differential gene expression (nominal p < 0.05) at 24 h and 206 transcripts at 48 h; 85 and 278 transcripts were differentially expressed at 24 and 48 h, respectively, in ZIKV-exposed mDCs (relative to noninfected control mDCs). Overall, these transcriptional changes were substantially weaker than the 886 and 2372 transcripts noted to be differentially expressed in vivo in pDCs and mDCs from ZIKV-infected patients, respectively (Fig. 4a). Notably, a computational analysis

of DEGs from in vitro infection experiments confirmed the strong activation of IFN-I-dependent immune activity in pDCs that we previously observed in transcriptional signatures from ZIKV-infected patients; in contrast, a more modest activation of IFN-I-dependent immune activity was observed in gene expression signatures from in vitro-infected and patient-derived mDCs (Fig. 4b, Supplementary Fig. 2b, and Supplementary Data 6). Differential responses of pDCs and mDCs to ZIKV infection were also supported by an algorithm-based analysis of transcripts exhibiting the highest level of connectivity among DEGs from in vitro-infected and from ZIKV-infected patient cells. These computational studies further identify components of IFN signaling pathways, such as JAK/STAT and mitogen-activated protein kinases, as critical hubs for the transcriptional response to ZIKV infection in pDCs (Fig. 4c and Supplementary Fig. 2c), consistent with previous studies[24–27].

**Social network analysis of immune cells**. Interactions and mutual connections between different immune cell subsets may play an important, but yet underappreciated, aspect of the human immune response to ZIKV infection. To address this, we used an unbiased gene clustering approach to define modules of co-regulated genes with correlated transcriptional activity within each of the analyzed cell populations. As demonstrated in Fig. 5a, Supplementary Fig. 3a, and Supplementary Data 7, we identified distinct patterns of gene modules within each cell population that were clearly distinguishable and almost mutually exclusive between ZIKV-infected patients and control persons, and were most obvious in pDCs from ZIKV-infected individuals. Remarkably, a correlation analysis demonstrated close statistical linkages between individual gene modules from different cell populations within the ZIKV-infected patients, suggesting functional connections between distinct cell compartments during ZIKV infection (Fig. 5b). Associations between these gene modules were strictly segregated between ZIKV-infected patients and uninfected patients (Fig. 5b and Supplementary Fig. 3c), suggesting that ZIKV infection profoundly alters and interferes with physiological multi-compartment immune networks. Notably, transcriptional modules in pDCs displayed multiple diverse connections with individual modules identified in alternative cell subsets, consistent with a central and highly interactive role of pDCs in regulating and orchestrating immune defenses against ZIKV (Fig. 5b). To investigate associations between gene modules from a functional perspective, we determined predicted biological pathway enrichments across correlated gene modules. These studies identified the mTOR and MAP kinase pathways, and to a lesser degree, the integrin signaling pathway, as underlying features of the statistically associated modules. Moreover, IFN-I-mediated antiviral immune defense emerged as an integrative immune signature linking distinct gene modules from different cell populations (Supplementary Fig. 3b). We subsequently explored linkages among individual cell populations on the level of individual genes that were statistically associated across multiple cell populations, as shown in Fig. 5c. Strikingly, this topological analysis indicated a high degree of gene co-expression among different gene modules observed in pDCs and other cell compartments, suggesting that a substantial number of genes from pDCs participate in interconnected, multi-compartment immune response programs to ZIKV infection. In addition, pDC genes co-expressed with transcripts in other cell types were frequently involved in a diverse spectrum of functional pathways with important roles in regulating of host cell behavior and immune defense mechanisms, including the HIPPO, RhoA, and inositol trisphosphate signaling cascades (Fig. 5d and Supplementary Data 8). Remarkably, statistical linkages between individual gene expression intensities were particularly obvious for transcripts expressed in pDCs and B cells, and a considerable number of genes with such correlated gene expression patterns in both cell subsets were predicted to be involved in critical B-cell functions, such as B-cell receptor signaling and phosphoinositide 3-kinase signaling in B lymphocytes. Moreover, the lower frequency of IgD and IgM transcripts and the higher frequency of IgG, IgA, and IgE transcripts in B cells of ZIKV-infected patients compared to uninfected donors (Supplementary Fig. 4a) supports the idea of a higher maturation and differentiation of B cells following ZIKV infection. Together, these data suggest that pDCs form multi-directional collaborative interactions with alternative immune cell subsets during ZIKV infection.

**Integrative functional roles of pDCs in ZIKV immune defense**. We subsequently conducted functional in vitro infection experiments to further explore immune responses of pDCs against ZIKV. We observed that during in vitro culture, experimental depletion of pDCs strongly enhanced the susceptibility of PBMCs to ZIKV infection, while profoundly reducing IFNA mRNA levels in response to ZIKV infection, emphasizing the critical role of pDC-dependent type I IFN responses for effective human immune defense against ZIKV (Fig. 6a, b and Supplementary Fig. 5b). Of note, inactivation of ZIKV by UV light markedly reduced IFNA mRNA expression in ZIKV-exposed pDCs, indicating that the observed effects were unrelated to nonspecific contaminants in viral stocks (Supplementary Fig. 5a-c). Moreover, following in vitro infection, pDCs expressed five- to tenfold higher levels of the co-stimulatory molecule CD86, likely reflecting activation of potent cell-intrinsic viral immune recognition pathways in pDCs (Fig. 6c). In contrast, B cells displayed only twofold higher levels of CD86 following ZIKV infection, whereas no CD86 upregulation at all was noticed in monocytes and mDCs (Fig. 6c). Unlike T and NK cells, B cells had the ability to increase surface expression of the early activation marker CD69 in response to ZIKV infection of total PBMC; however, this upregulation was significantly diminished after experimental depletion of pDCs, suggesting that functional connections between pDCs and B cells are necessary to effectively activate B cells following ZIKV exposure (Fig. 6d and Supplementary Fig. 5d). Using co-culture experiments with purified B cells and pDCs, we confirmed that B-cell activation following ZIKV infection was strongly dependent on cellular interactions between B cells and pDCs, and almost completely abrogated by antibodies blocking type I IFN and by physical separation of pDCs and B cells using transwell co-culture systems (Fig. 6e, Supplementary Figs. 4c and 5e). These observations suggest that upon ZIKV exposure, pDCs can effectively activate B cells in their immediate physical proximity through secretion of type I IFN, consistent with prior findings in the context of alternative flaviviruses[27–29]. Together, these results highlight the distinct, dual ability of pDCs to generate and orchestrate both cell-intrinsic and collaborative immune responses against ZIKV.

## Discussion
ZIKV infection causes a transient acute illness that in rare cases can be associated with severe neurological disease manifestations, specifically during fetal development. Although previously underappreciated, it is now clear that ZIKV can induce a strong immune response in humans that includes both innate and adaptive immune elements, and successfully contains viral infection within a limited time in the majority of cases. However, it is increasingly evident that ZIKV has developed multiple strategies to reprogram, antagonize, or corrupt host immunity in favor of the virus. In this work, we conducted detailed

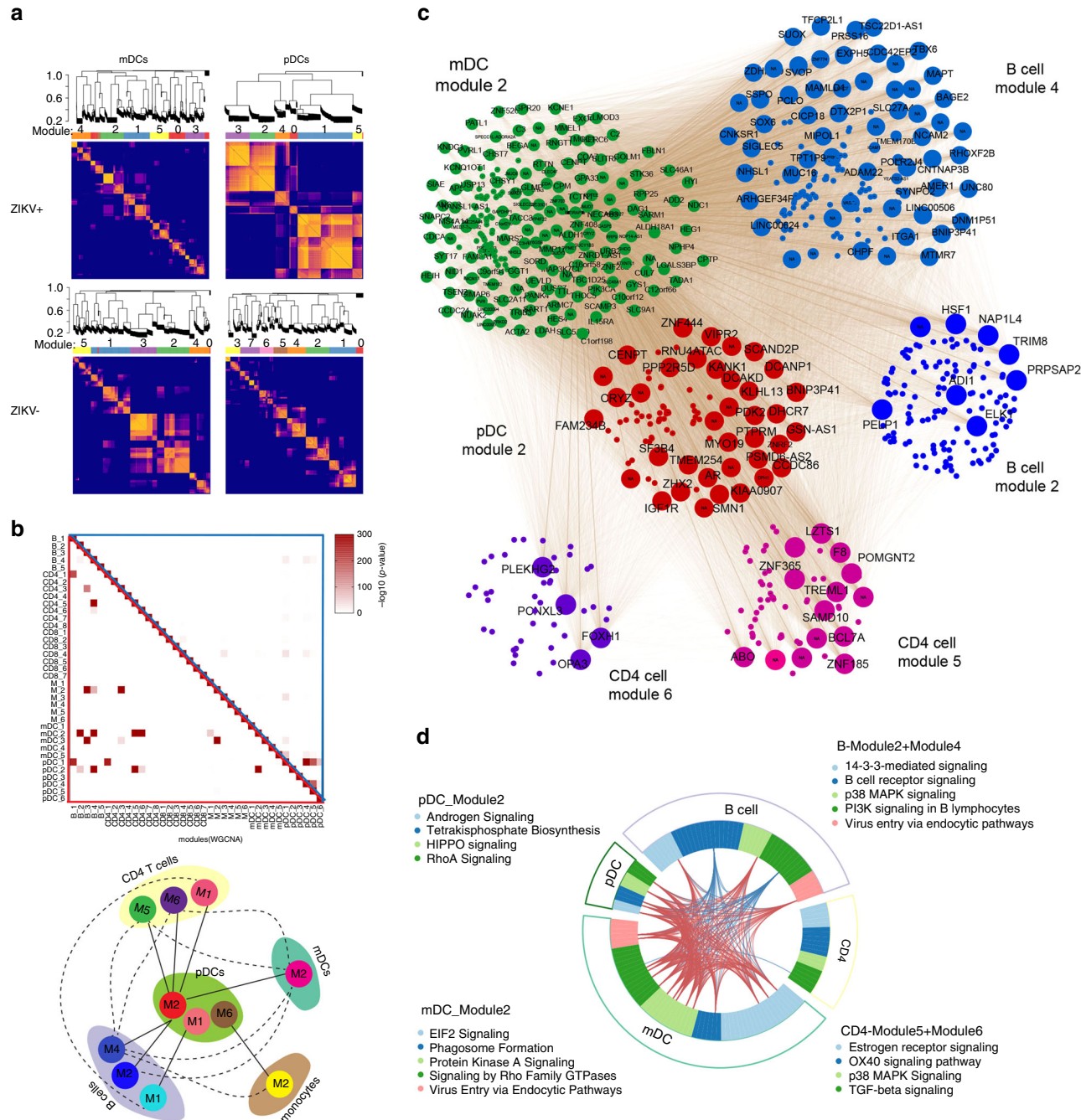

**Fig. 5 Gene module correlation analysis highlights distinct but integrative role of pDCs for antiviral immune defense against ZIKV. a** Heatmaps highlighting modules of transcripts with correlated gene expression patterns in indicated immune cells subsets from ZIKV-infected (ZIKV+) and noninfected individuals (ZIKV−). **b** Uupper panel: Correlation analysis of individual gene modules from indicated cell compartments defined in ZIKV-infected patients, applied to transcriptional profiling data from ZIKV-infected patients (red-framed triangle) or from ZIKV-noninfected individuals (blue-framed triangle). Lower panel: Graphic illustration of interconnected gene modules from different immune cell subsets of ZIKV-infected patients. **c** Network of correlated genes within B-cell module 2 and 4, CD4 module 5 and 6, mDC module 2 and pDC module 2. Significant correlations between genes expressed in different cell subtypes are represented by gray lines. Size of the genes symbols indicates the number of connections. **d** Circos plot highlighting co-expressed genes, categorized according to cell types, gene modules, and predicted functional annotation.

transcriptional profiling studies in all major immune cell subsets from a group of naturally-infected individuals, which to our knowledge represents one of the most comprehensive investigations of the host response to ZIKV infection in humans.

While demonstrating complex, cell subtype-specific variations in the transcriptional response to ZIKV infection, our results indicate a remarkably distinct role of pDCs in the human immune response to ZIKV. In particular, we show that

pDCs differ from other cell subsets by upregulating a cluster of antiviral response genes, including two members of the IFN-induced transmembrane (IFITM) protein superfamily, *IFITM3* and *TMEM2*, which have been associated with inhibition of ZIKV, influenza virus, and HBV replication in prior work[23,30–32]. Moreover, pDCs from ZIKV-infected patients appeared to downregulate key viral dependency genes, such as the critical ZIKV entry receptor *AXL*, which has a known role

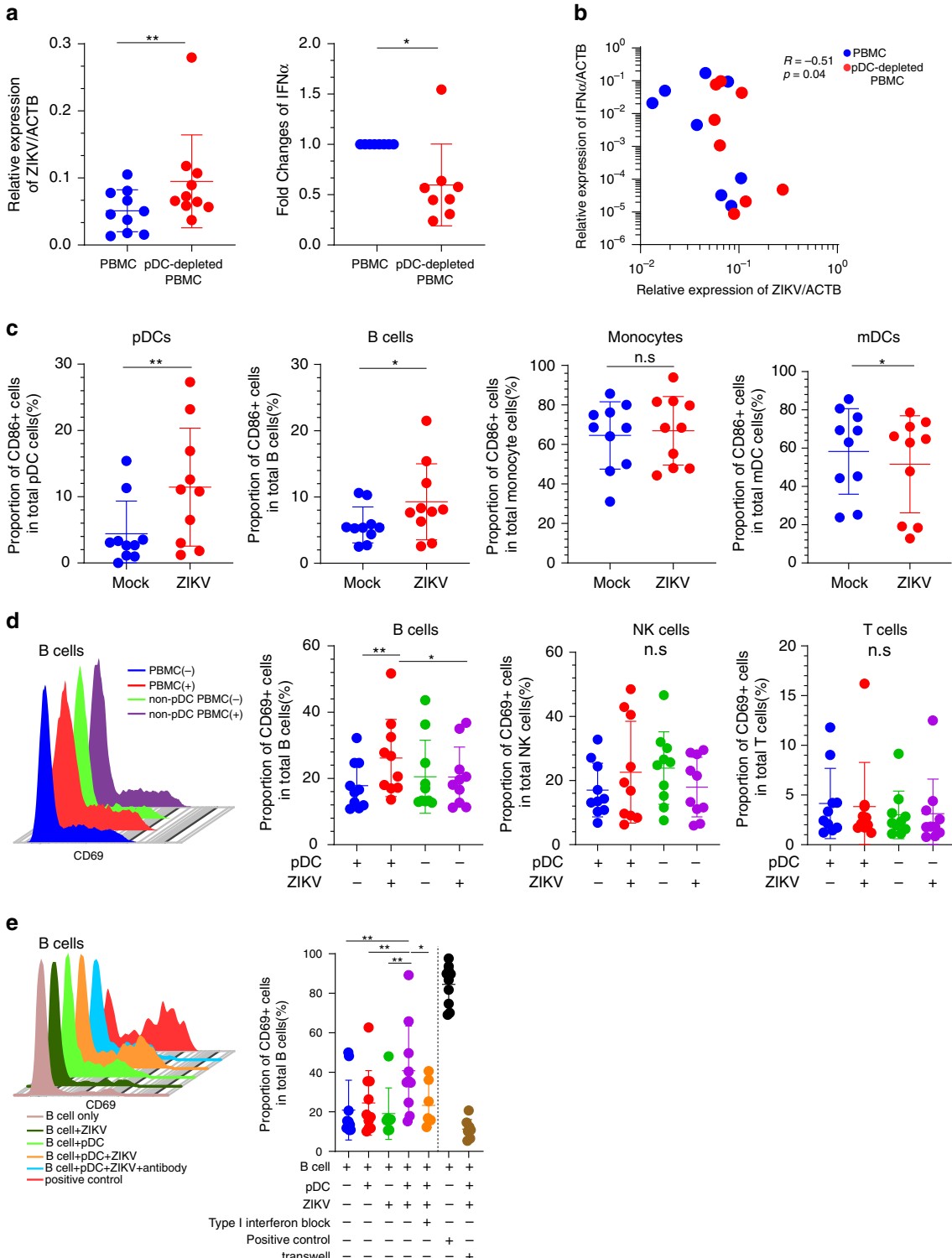

in regulating type I IFN responses[33]. These combined changes seem to lead to a profoundly reduced susceptibility of pDCs to ZIKV infection, in striking contrast to the high susceptibility of mDCs to ZIKV described previously by us and others[15,16]. At the same time, we observed multiple genes and gene modules with closely correlated gene expression patterns in pDCs and alternative cell subsets, pointing to a role of pDCs in integrating, coordinating, and orchestrating networks of antiviral immune defense. Together, these observations suggest that pDCs oscillate between the dichotomous role of a discrete

outlier and an integral component of antiviral immune defense networks against ZIKV.

A key finding of this study is the markedly different cell-intrinsic immune response of pDCs to ZIKV, relative to essentially all other analyzed immune cell types. We propose that this distinct transcriptional reaction to ZIKV infection in pDCs is likely a result of the discrete innate pathogen recognition machinery in pDCs that allows for improved ZIKV RNA sensing in this particular cell subset. Indeed, pDCs are known to express TLR7 and TLR9, both of which can effectively trigger

**Fig. 6 Functional roles of pDCs in the human immune response to ZIKV. a** *ZIKV* RNA (*n* = 10 biologically independent samples) and *IFNA* mRNA (*n* = 8 biologically independent samples) expression in total and pDC-depleted PBMCs at 24 h following in vitro infection with ZIKV. **b** Correlation between *ZIKV* RNA expression and corresponding *IFNA* mRNA expression in total PBMC and pDC-depleted PBMC. Spearman's correlation coefficient is indicated. Cumulative data were analyzed using generalized estimated equations adjusted for repeated measures. **c**. CD86 surface expression in indicated cell populations at 24 h following in vitro infection with ZIKV (*n* = 10 biologically independent samples). **d** Expression of the early activation marker CD69 on B, T, and NK cells at 24 h after in vitro infection of total or pDC-depleted PBMCs. Uninfected control cells are shown for comparative purposes (*n* = 9 biologically independent samples). **e** CD69 surface expression on isolated B cells following exposure to ZIKV, in the presence or absence of co-cultured autologous pDCs (*n* = 9 biologically independent samples). Results of culture conditions with type I interferon blocking antibodies, with transwell co-culture systems separating pDCs and B cells or with TLR9 ligands (ODN 2395; positive control) are also shown as controls. Data are presented as mean ± SD. Statistical significance between the different subsets was tested using two-sided Wilcoxon matched-pairs signed-rank tests for **a–c**. Statistical significance between the different subsets was tested using a Friedman test with post-hoc Dunn's test for **d**, **e**. Correlation between type I interferon and *ZIKV* RNA were analyzed using generalized estimated equations adjusted for repeated measures for 6b. n.s: not significant; *$p < 0.05$, **: $p < 0.01$.

cell-intrinsic type I IFN responses. Moreover, recent studies have shown that pDCs have the capacity to interact with cells infected by multiple flaviviruses (dengue, hepatitis C, and ZIKVs) via the αLβ2 integrin/ICAM-1 cell adhesion molecules, and induce an antiviral response by forming an "interferogenic synapse"[27]. In contrast, mDCs seem to primarily rely on the RIG-I pathway for the detection of RNA viruses[34,35]. Although RIG-I is clearly able to sense ZIKV, numerous studies have identified ZIKV-induced disruption of downstream signal transduction after cytoplasmic immune recognition in mDCs, specifically at the level of STAT1 phosphorylation and proteasomal degradation of STAT2 (refs. [19,24,36,37]).

Even though the critical role of pDCs in antiviral immune defense is generally well-recognized, specifically in the context of animal experimentation studies and in vitro experimental settings, there is a considerable level of uncertainty about the functions of pDCs in infected humans during natural disease courses. In the context of ZIKV infection, no clear changes in pDC phenotype were observed in cells derived from ZIKV-infected female patients[38]. Moreover, pDCs represented the earliest responder cell population in male pigtail macaques (PTM) experimentally infected with ZIKV[26], demonstrating a substantial numerical increase in circulating blood within 1–5 days after virus inoculation; this was associated with upregulation of activation markers on the cell surface of pDCs. However, female PTM failed to demonstrate a notable phenotypic change in pDCs, indicating notable sex differences in the pDC-mediated antiviral immune response[26]. Our work, which was conducted exclusively in women and relied on detailed, unbiased, and comprehensive transcriptional signature evaluations, clearly indicated a profound and distinct response of pDCs to ZIKV infection.

Although recent reports suggest persisting reservoirs of ZIKV-infected cells in rare cases, namely in the genitourinary tract[39], most ZIKV-infected patients manage to clear the infection after several days or weeks. This clearance seems to be primarily linked to an effective neutralizing antibody response[40], and to IFN-γ secreting ZIKV-specific CD4 T cells circulating in peripheral blood[41]. The protective effects of antibodies to ZIKV infection also have been confirmed in mice and non-human primate models[40,42,43]. In addition, several studies in animal models have suggested that protection against ZIKV infection can be achieved through vaccines that induce protective antibody responses[44–48]. We have shown here that pDCs seem to represent the major antigen-presenting cell population that is effectively activated and engaged during natural infection in humans. This activation, and the associated production of type I IFN, is likely to play an important role in the recruitment and evolution of the adaptive immune response. In this regard, our observation of statistically correlated gene expression modules between pDCs and T/B cells suggest a critical functional link between pDCs and adaptive effector cell responses, and provides empirical evidence to

support the notion that pDC activation may play a pivotal role for priming ZIKV-specific T- and B-cell immunity. In the context of ZIKV infection, studies have suggested that immune protection can be mediated by B cells through production of neutralizing antibodies[49,50], and it is tempting to hypothesize that pDCs may have a distinct function for inducing and fine-tuning such ZIKV-specific B-cell responses. Correspondingly, studies in rotavirus and influenza virus infection have shown that pDCs were necessary and sufficient for generating antibody-secreting plasma cells through type I IFN and cytokine responses[51,52]. A more systematic analysis of how innate pathogen recognition and immune defense pathways intertwine with the evolution of effective adaptive immune responses would likely require complex, longitudinal immune monitoring studies of ZIKV-infected individuals within short intervals after viral infection, an endeavor that cannot be easily accomplished due to logistical challenges in identifying study individuals willing to undergo such testing. Nevertheless, a precise profiling of immune networks during naturally acquired ZIKV infection has the potential to identify previously-unrecognized immune interactions occurring during a successful antiviral human immune response.

## Methods

**Study participants**. ZIKV-infected study participants were recruited from the Massachusetts General Hospital and the Brigham and Women's Hospital (Boston, MA). PBMC samples were used under protocols approved by the Partners Human Research Committee, the local institutional review board (protocol number 2016P000319). Clinical and demographical characteristics of study patients were described previously[16] and are summarized in Supplementary Table 1. Study patients gave written informed consent to participate in accordance with the Declaration of Helsinki.

**Flow cytometry and cell sorting**. PBMCs were stained with antibodies from Biolegend against CD3 (clone OKT3,1:50), CD4 (clone OKT4,1:50), CD8 (clone SK1,1:50), CD19 (clone HIB19,1:50), CD56 (clone HCD56,1:50), CD16 (clone B73.1,1:50), CD14 (clone HCD14,1:50), HLA-DR (clone L243,1:33), CD11c (clone 3.9,1:33), CD123 (clone 6H6,1:33), and Blue viability dye (BVD, Life Technologies). The cells were then washed with phosphate-buffered saline (PBS) with 0.5% fetal bovine serum (FBS) and subjected to live cell sorting of CD4 T cells (100,000 cells with CD3⁺ CD4⁺ phenotype), CD8 T cells (100,000 cells with CD3⁺ CD8⁺ phenotype), monocytes (100,000 cells with CD3⁻ CD19⁻CD14⁺ phenotype), NK cells (100,000 cells with CD3⁻ CD19⁻ CD14⁻CD56⁺ phenotype), B cells (100,000 cells with CD3⁻ CD14⁻ CD19⁺ phenotype), pDCs (about 2,000 cells with CD3⁻ CD14⁻ CD19⁻CD56⁻HLA-DR⁺ CD11c⁻ CD123⁺ phenotype), and mDCs (about 4,000 cells with CD3⁻ CD14⁻ CD19⁻ CD56⁻ HLA-DR⁺ CD11c⁺ CD123⁻ phenotype) using a FACS Aria II cell sorter (BD Biosciences) at 70 pounds per square inch. The gating strategy is highlighted in Supplementary Fig 1a; numbers of cells sorted for each cell population are shown in Fig. 1c. Cell sorting was performed by the Ragon Institute Imaging Core Facility and resulted in the isolation of these 7 subsets with the defined phenotypic characteristics at >95% purity.

**Transcriptional profiling**. Total RNA was extracted from the 7 seven populations using a commercial PicoPure RNA Isolation Kit (Thermo Fisher Scientific). RNA-Seq libraries were generated as described in a previous study[53], using at total of 10 ng of RNA and external ERCC RNA spike-in control as input material for all cell subsets. Briefly, whole transcriptome amplification and tagmentation-based

library preparation was performed using the SMART-seq2 protocol, followed by sequencing with a 75-cycle kit on a NextSeq 500 instrument (Illumina, CA). The quantification of transcript abundance was conducted using the RSEM software (v1.2.22) supported by the STAR aligner software (STAR_2.4.2a). The raw reads were aligned to the Hg38 human genome database and the ZIKV sequence (ZIKV strain PRVABC59, GenBank: KU501215.1)[54–56].

**ZIKV production and titration**. ZIKV Strain PRVABC59 was isolated from a patient who traveled to Puerto Rico in 2015 (ref. [57]). The virus was produced using the Vero cell line and was kindly provided to us by the US Centers for Disease Control in February 2016. The virus was propagated only once in our lab using the C6/36 cell line. Viral stocks were titrated by flow cytometry on LLC-MK2 cells (ATCC) using the PE-conjugated 4G2 antibody (Novus Biologicals) which recognizes a conserved epitope on the E protein of flaviviruses. The virus stocks were then stored at −80 °C in Dulbecco's modified Eagle medium with 10% FBS. When indicated, ZIKV was exposed to UV light (wavelength 250-270 nm) at a distance of ~75 cm in a Class II plus Biosafety cabinet (BSL2+) at room temperature.

**In vitro infection experiments**. PBMCs from healthy individuals were infected with ZIKV and UV-inactivated ZIKV at a multiplicity of infection (MOI) of 1. The cells were incubated for 2 h at 37 °C. Next, the inoculum was removed, and the cells were washed two times with 10 ml of PBS. Subsequently, culture medium was added to each well and the cells were incubated at 37 °C and 5% $CO_2$ for the duration of the experiment. Different cell subsets were then sorted by FACS as described above. When indicated, PBMCs isolated from healthy donors were subjected to pDC depletion using CD304 microbeads (Miltenyi Biotec, Germany), followed by infection of total PBMCs or pDC-depleted PBMCs with ZIKV at a MOI of 1. Two hours after infection, PBMCs were washed twice, incubated at 37 °C and 5% $CO_2$ for 24 h, and subjected to RNA extraction or stained with antibodies against CD3, CD19, CD16, CD56, CD14, HLA-DR, CD11c, CD123 (see above-listed antibody clones), CD69 (clone FN50, 1:40), and CD86 (clone IT2.2,1:40) for subsequent flow cytometric analysis.

**Computational data analysis of RNA-Seq data**. To detect DEGs, DESeq2 implemented in the Bioconductor/R-project package was used to calculate FDR-adjusted p-values. IPA was used to functionally categorize DEGs. IPA is a software program, which can analyze the gene expression patterns using a build-in scientific literature based database[58] (QIAGEN Inc., https://digitalinsights.qiagen.com/products-overview/discovery-insights-portfolio/analysis-and-visualization/qiagen-ipa/). Gene co-expression networks were constructed and analyzed using the WGCNA algorithm;[59] power values for soft thresholds were determined automatically by the package. The robustness of the procedure was ensured by using the biweight midcorrelation function to quantify the correlation between genes, and by using a signed hybrid network using the topological overlapping matrix method for adjacency metrics, as recommended by the software author. Gene modules were identified by the dynamic tree cut algorithm. Correlation coefficients between genes from different cell types were calculated. Viral dependency genes were identified according to a previously described list[21]. ISGs were downloaded from the Interferome v2.0 database[60]. Networks of hub genes were constructed based on the STING interactome database[61] (confidence score cutoff at 900) and visualized using Networkanalyst[62]. GO terms of each hub gene were defined by GeneCards[63]. The ontology of DEGs was determined by gene set enrichment analysis[64,65] or the WebGestalt analysis approach (WEB-based Gene SeT AnaLysis Toolkit, http://www.webgestalt.org/)[66]. DEGs in each subset were tested for enrichment with viral dependency genes[21] and ISGs[60], using the public available software GSEA (www.broadinstitute.org/gsea)[64,65] with default settings.

**Small interference RNA experiments**. Selected siRNA molecules (On-TARGET plus SMARTpool, Dharmacon) were transfected using the DOTAP liposomal transfection reagent (Roche, Switzerland), as described in a previous study[67]. Briefly, human pDCs were isolated from healthy donors by negative selection using the plasmacytoid Dendritic Cell Isolation Kit II (Miltenyi Biotec, Germany), with a purity of >95%, as determined by flow cytometry (Supplementary Fig. 1c). Isolated pDCs were seeded at $10^5$ cells/100 μL in 96-well plates and incubated at 37 °C. siRNAs were diluted in PBS for a final concentration of 160 nM and mixed with 1:1 with DOTAP (Roche). The mix was incubated at room temperature for 15 minutes and then added to cells and incubated for 24 h without medium change. After 24 h post transfection, a small cell sample was collected to check the gene silencing efficacy by qPCR; the remaining cells were infected with ZIKV at a MOI of 1. At defined time points after infection, total ZIKV RNA was quantified by quantitative RT-PCR (qRT-PCR) as described in our previous study[16].

**Tissue culture experiments**. For co-culture experiments, pDCs and B cells were individually isolated from healthy volunteers, using negative immunomagnetic selection procedures (Miltenyi Biotech). Subsequently, pDCs and B cells were cultured in 96-well plates at pDC: B-cell ratios of 0.4:1, and infected with ZIKV at a MOI of 1. At defined time points after infection, B cells infected in the presence or absence of pDCs were subjected to immunophenotyping by flow cytometry, using the following antibodies: CD19 (FITC, clone HIB19), CD123 (BV510, clone 6H6),

CD303 (PerCP, clone 201A), CD69 (BV605, clone FN50), in addition to staining with BVD for isolation of live cells. When indicated, a human type I IFN neutralizing antibody mixture (PBL Assay Science) which effectively neutralizes biological activity of human IFN-α, IFN-β, IFN-Ω, IFN-κ, and IFN-ε was added. Class C CpG oligonucleotides (ODN 2395, InvivoGen), a TLR9 ligand, served as a positive control to activate B cells. In selected cases, isolated B cells ($1 \times 10^5$ /well) and pDC ($0.4 \times 10^5$) were co-cultured using the upper and lower compartments of a transwell culture system (0.4 μm, Costar, Wilmington, MA), respectively; co-culture of isolated T cells and pDC under similar conditions was used as a control. All flow cytometry data was analyzed using FlowJo v10 (FlowJo, USA).

**qRT-PCR for selected host genes**. Total RNA was extracted using PicoPure RNA Isolation kits, followed by cDNA generation with random hexamer primers (Thermo Scientific, USA) and the Superscript III kit according to standard procedures. The qRT-PCR was performed using the QuantiTect Probe RT-PCR kit (Qiagen, USA) according to the manufacturer's instructions. qPCR reactions for detection of IFIH1, TMEM2, IFITM3, IFNA, IFNB, and ACTB (see primer sequences in Supplementary Table 2) were performed using the SsoAdvance Universal SYBR Green Supermix (BioRad, USA). Data were normalized to β-actin expression determined with commercial primers (Hs0160665_g1, Life Technologies, USA).

**ZIKV qRT-PCR**. Strand-specific ZIKV qRT-PCR was performed to evaluate negative-strand ZIKV RNA which is a marker of ongoing viral replication[16,22,68,69]. Briefly, isolated RNA was reverse transcribed using the superscript III kit (Invitrogen, USA), with a ZIKV-specific forward primer for negative-strand RNA and random hexamers for total ZIKV RNA. cDNA was then amplified with ZIKV Env- specific primers and probes (Supplementary Table 2). The relative amount of total and negative-strand viral RNA was calculated using the $2(-\Delta\Delta^{CT})$ method using ACTB (Hs0160665_g1) as the internal control for normalization. All qRT-PCR and the qPCR reactions were run on an Applied Biosystems Vii7 real-time PCR system (Applied Biosystems, USA). For detection of ZIKV RNA ex vivo in patient-derived samples, the exact viral copy number was calculated using ZIKV RNA standards. To generate standards, a previously described protocol was used[22]. Briefly, total viral RNA from ZIKV-infected C6/36 cells (ATCC) was purified using a PicoPure RNA Isolation kit following the manufacturer's protocol. A standard RT-PCR was then carried out by using primers containing the T7 promoter sequence (5′-TAATACGACTCACTATAG-3′). The PCR product was used to generate ZIKV RNA fragments by in vitro transcription using a MAXIscript kit (Ambion, Austin, TX).

**Statistics**. Statistical significance between the different subsets was tested using Mann–Whitney U-tests or Wilcoxon matched-pairs signed-rank tests. When appropriate, statistical analysis was corrected for multiple comparisons using a Friedman test with post-hoc Dunn's test. Correlation between type I IFN and ZIKV RNA were analyzed using generalized estimated equations adjusted for repeated measures. NS, not significant; *$p < 0.05$, **$p < 0.01$, ***$p < 0.001$, ****$p < 0.0001$. Experiments were conducted once with each described sample unless indicated otherwise in the text. Statistical significance was analyzed in GraphPad Prism v7, RNA-seq data analysis was performed in R v3.5.2 and graphical network were plotted using Cytoscape v3.7.2.

**Reporting summary**. Further information on research design is available in the Nature Research Reporting Summary linked to this article.

## Data availability

Source data for Figs. 1–6 and Supplementary Figs. 1–5 are provided with the paper. RNA-Seq data have been deposited to the NCBI GEO and are available under accession number GSE132228. Hg38 human genome database and Interferome v2.0 database[60] were used in this study. All other data are available from the corresponding author upon reasonable request.

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

## Acknowledgements

X.G.Y. is supported by NIH (grants AI078799, AI089339, HL121890, AI098484, HL126554, AI116228, and AI087452) and by a Sundry fund from the Ragon Institute of MGH, MIT, and Harvard.

## Author contributions

X.S., S.H., M.L., and X.G.Y. developed the research idea and study concept, designed the study, and wrote the manuscript. X.G.Y. supervised the study. X.S. and S.H. designed and conducted experiments. J.E.B. provided technical support for qPCR and qRT-PCR experiments. C.G., Z.O., X.S., and S.H. conducted the statistical and bioinformatic analysis for RNA-Seq data. M.L., K.A., A.C., S.Y., P.S., and E.S.R. provided samples from ZIKV-infected patients. All authors critically reviewed the manuscript.

## Competing interests

The authors declare no competing interests.
