## [Peer Review File · Nature Communications]

Reviewers' comments:

Reviewer #1 (Remarks to the Author):

Plasmacytoid dendritic cells (pDC) are specialized in rapid and high level production of the cytokines types I and III interferons (IFN-I/III) in response to viral infections, without being infected, through uptake, endosomal routing and sensing by Toll-like receptors 7 or 9 of viral nucleic acids. pDC have been shown to be broadly resistant to viral infections. In contrast, other mononuclear phagocytes including monocytes, macrophages and type 2 conventional dendritic cells (cDC2) are more susceptible to viral infections, and when these cells are infected, their functions can be inhibited or hijacked by viral immune evasion genes. Indeed, many viruses harbor genes which can inhibit or blunt the production of IFN-I/III by infected cells. Since IFN-I/III responses are critical for host resistance to viral infections both in mice and humans, it was therefore inferred that pDC have a critical beneficial role in antiviral defense by promoting the production of high levels of the cytokines in the face of the viral immune evasion mechanisms acting in infected cells. However, in mice, data demonstrating a non-redundant and critical role of DC for antiviral defense are scarce. Moreover, humans bearing primary immune deficiencies leading to the loss of pDC IFN-I/III production do not show enhanced susceptibility to viral infections. Thus, whether pDC play a unique and critical role promoting host defense during viral infections is still a matter of debate in mice and even more in humans. More generally, to the best of my knowledge, how pDC are activated as compared to other mononuclear phagocytes in vivo during an acute viral infection in humans has not been studied yet, in part due to the difficulty in diagnosing the disease and harvesting blood samples early enough after infection. The current paper takes advantage of unique and rare blood samples harvested in three patients during acute infection with the zika Virus (ZIKV) in order to address these questions. Specifically, the activation state of the pDC of these patients were characterized ex vivo by gene expression profiling, as compared to pDC from matched healthy controls as well as with other immune cell populations isolated from the same blood samples, including two other types of mononuclear phagocytes, monocytes and so-called myeloid dendritic cells (mDCs), as well as B lymphocytes, natural killer (NK) cells, CD4+ T cells and CD8+ T cells. A variety of computational analyses were then performed on the gene modules found to be differentially expressed between at least on cell type from the patients and its counterpart from healthy controls, to infer the associated biological processes, their molecular regulation and their potential relevance to the orchestration of antiviral immunity and to the defense of the host.

This paper represents a major "tour de force" by combining the study of very unique, precious and interesting clinical samples with cutting-edge technological approaches including micro-bulk RNA sequencing on 7 different immune cell populations and deep computational analyses of the data. This study will likely attract considerable attention in the broad community of researchers interested in antiviral immunity and in the biology of pDC.

The major claims of the paper are the following.

Claim #1) Contrary to other mononuclear phagocytes (monocytes and mDC), pDC are resistant to CHIKV infection, in part due to their downregulation of ZIKV dependency genes, and also due to their unique upregulation of a specific set of restriction factors.

Claim #2) Contrary to other mononuclear phagocytes (monocytes and mDC), pDC mount strong IFN-I responses to ZIKV infection, likely due to their ability to sense the virus through TLR7, without being infected and thus without being affected by the immune evasion genes expressed in infected mononuclear phagocytes.

Claim #3) Social network analysis of immune cells following ZIKV infection highlight a unique and central role of pDC in the orchestration of antiviral immunity.

A major limitation of the paper is that it stops at the stage of inferences from the analysis of the gene expression profiling, without formulating precise enough hypothesis and testing them experimentally. Hence, additional experiments are required to increase the significance/impact of the study by experimentally testing some of functions proposed to be unique for pDC and to promote antiviral defense.

Moreover, a number of conclusions in the paper are overstated, either due to inconsistencies in the use of the results of the statistical analyses performed to identify differentially expressed genes, or because of insufficient quantitation of the differences reported between the different cell types studied. Hence, additional analyses or experiments, and some rewriting, are required to strengthen or modulate some of the conclusions of the paper.

Finally, the main conclusion about the unique role of pDC should also be modulated, by discussing them in a broader context, considering that the authors have not studied another type of DC known to be critical for antiviral defense, namely type 1 cDC (cDC1). Indeed, cDC1 share with pDC their broad resistance to viral infections [PMID: 28783704], the ability to produce IFN- β and IFN- γ without being infected (in response to TLR3 triggering) [PMID: 20975040], and have been clearly and indisputably demonstrated in mice to be absolutely critical to promote protective antiviral innate and adaptive immune responses [PMID: 19008445; 22505455; 22505458; 26296422; 26297566; 27692611; 28190711; 30409884]. Hence, a number of the characteristics of in vivo pDC responses to acute ZIKV infection in humans may not be unique to this cell type but largely shared by cDC1, with both overlapping and complementary roles of these two cell types in the global orchestration of protective antiviral immune responses.

MAJOR POINTS.

Major point #1) More precise hypotheses should be formulated from the computational analyses performed, and then tested experimentally.

1-1) The authors should test whether the knock-down of IFIH1, IFITM3 or TMEM2, or enforced ectopic expression of EMC7, AXL or IDO1, in pDC increases their susceptibility to ZIKV infection.

1-2) The authors should examine how pDC depletion from healthy donor PBMCs before their exposure to ZIKV alters the transcriptional reprogramming of other immune cell types, in particular B cells, to experimentally test their hypothesis of a central role of pDCs in immune defense during ZIKV acute infection through their establishment of immunological circuits and other cells, in particular with B cells to promote their activation.

Major point #2) A number of conclusions in the paper are overstated due to insufficient rigor in data description and interpretation.

A number of results that are not significant according to the figures or the data tables are discussed in the main text as if they are. This is not appropriate and must be corrected. Please find below a non-exhaustive list of examples.

2-1) Line 204, contrary to what is written in the main text, the direction of the modulation of the "antigen presentation" canonical pathway in B cells and in mDCs is undetermined (Figure 2a) in the computational analysis, and thus not documented as inhibited.

2-2) It is not clear how the authors have selected the terms used to describe the results of the functional annotations of the DEGs (Figure 2b), since "Type I IFN immune responses" or "effector immune cell recruitment" are not used in the functional annotation terms included in the corresponding figure and supplemental table. Please use the ontology directly provided from the database mined for the computational analysis. If an additional layer of interpretation can be added upon further mining of the gene lists, to refine the nature of the biological process at play, please justify precisely how this was achieved.

2-3) Line 215-216, it is not true that the involvement in transcriptional regulation was markedly less obvious in alternative cell populations than in pDCs for all of the cytokine upstream regulators listed. For example, Figure 2c clearly shows as strong a significance for IFNbeta, IFNA2, Interferon alpha in B cells and CD8+ T cells than in pDC, and this is also the case for IL15, IL3, EBI3, TNFSF10, IFNL1 and OSM, etc... The difference stand out mostly for the upstream regulators that are central to the heatmap, including CCL5, TNF and CSF1.

2-4) Lines 235-236. When taking into account the statistical analysis, it is not true that EMC7, AXL and IDO1 were downregulated in pDC but upregulated in the other cell populations. The changes in expression of EMC7 and AXL in pDC are not significant. The only significant changes are: an up-regulation of AXL and IDO1 in mDC, and an up-regulation of EMC7 in B cells and monocytes. Of note, the data for IDO1 in pDC in the supplemental table is missing, which must be corrected. Rather, when looking at figure 3b, it appears that one of the characteristic of pDC response in acute ZIKV infection is their lack of significant upregulation of any of the known ZIKV dependency genes, whereas a number of these genes are upregulated in B cells, mDC and monocytes. A Gene Set Enrichment Analysis (GSEA) would be welcome here to compare each immune cell population between patients and controls, as well as to compare the pDC to each of the other cell populations in the patients, for enrichment in the gene set corresponding to the list of the known ZIKV dependency genes illustrated as Figure 3b. This would allow to better quantitate the global expression changes of the ZIKV dependency genes in patients as compared to controls, and the relative expression of this gene module between cell types in patients.

2-5) Lines 251-252. When taking into account the statistical analysis, it is not true that all the ZIKV restriction factors listed are upregulated in pDCs in contrast to alternative cell compartments. IFIH1, IFITM3 and TMEM2 were indeed significantly upregulated in pDCs and not significantly changed in B cells, monocytes and mDCs. However, IFI27 was significantly upregulated in mDCs and IFIT3 in monocytes but the expression of these genes did not change significantly in pDCs. The expression of TNFAIP2, TNFAIP3, IRF1 and OAS3 did not change significantly in pDCs. Here again, a GSEA would be welcome to better quantitate the global expression changes of the ZIKV restriction factor gene module in patients as compared to controls, and the relative expression of this gene module between cell types in patients.

2-6) Lines 253-255, the sentence "Moreover, a computational analysis of upstream regulators governing signatures of DEGs supported a critical role of type I IFN responses in pDCs from ZIKV-infected patients, in contrast to mDCs in which type I IFN signatures appeared to be reduced (Figure 3f)" needs to be supported by a more precise, quantitative, analysis of the data.

2-7) Lines 276-277, the sentence "no such activation of IFN-dependent immune pathways was seen in gene expression signatures from in vitro infected and patient-derived mDCs (Fig. 4c)." is an overstatement considering the positive z-score for "Interferon signaling" in mDC cells at 48 hours after in vitro stimulation, and ex vivo. Here again, a GSEA would be welcome to better quantitate the global expression changes of the ISG module in patients as compared to controls, and the relative expression of this gene module between cell types in patients.

Minor points.

Minor point #1) The main conclusion about the unique role of pDC should be modulated. First, on many aspects, B cells appear to behave to a large extent similarly to pDC. Second, the authors have not examined the behavior of cDC1 which may well share even more characteristics with pDC in their response to ZIKV acute infection. Hence, without further studies, the authors cannot state that their "results indicate a remarkably unique role of pDCs in the human immune response to ZIKV" as written in their title and

discussion.

Reviewer #2 (Remarks to the Author):

In this manuscript, the authors utilize a systems biology-based approach to understand the biological networks within immune cells isolated from ZIKV-infected individuals. As compared to other myeloid cells, plasmacytoid DCs (pDCs) displayed innate immune activation, characterized by the induction of type I interferon (IFN). While this study is interesting, the findings are largely descriptive with little to no biological validation. The study was difficult to read and the figures were largely a collection of heatmaps and "hairballs" of gene networks. In addition, there were a number of concerns with this study:

1. It is unclear if equal cell numbers were used across the transcriptional profiling studies. This is especially important with cells, such as s B cells.
2. This analysis is based on a single infection timepoint and this information is not readily found within the text of the results section. Rather than including a reference, this pertinent information should be included in this manuscript.
3. The canonical pathway analysis is rather confusing and misleading. Many of these pathways/biological functions can differ by only 1 or 2 genes. This is not reflected in the data presented in Figure 2. Moreover, precisely how are functional pathways predicted to be up or down regulated simply based on mRNA expression patterns? The authors need to provide biological validation of these data rather than predictive models.
4. It is not clear what the authors mean by "distinct role of pDCs" to ZIKV infection. pDCs are inherently a distinct cell population and are known to be major inducers of type I IFN. Moreover, it is also well known that flaviviruses do not infect pDCs.
5. The authors need to provide an in-depth analysis of ZIKV infection of mDCs and pDCs. Moreover, ZIKV is not a poly-adenylated virus. It is unclear how the authors were able to capture ZIKV RNA using an oligo-DT primed cDNA reaction for mRNA-seq. These results are not reliable and do not reflect ZIKV replication. What is the purity of the pDC population? Are pDCs infected in ZIKV-infected patients?
6. The findings that JAK/STAT pathways are critical hubs for ZIKV infection are not a new concept. Many flaviviruses are highly sensitive to type IFN and it is no surprise that replication/infection corresponds to IFN-related signatures.
7. Much of this manuscript is based on predictive computational modeling with little to no biological validation studies. It is not surprising that immune cells interact with each other. Several groups have shown the importance of pDCs in bridging innate and adaptive immune functions.

Reviewers' comments:

Reviewer #1 (Remarks to the Author):

Plasmacytoid dendritic cells (pDC) are specialized in rapid and high level production of the cytokines types I and III interferons (IFN-I/III) in response to viral infections, without being infected, through uptake, endosomal routing and sensing by Toll-like receptors 7 or 9 of viral nucleic acids. pDC have been shown to be broadly resistant to viral infections. In contrast, other mononuclear phagocytes including monocytes, macrophages and type 2 conventional dendritic cells (cDC2) are more susceptible to viral infections, and when these cells are infected, their functions can be inhibited or hijacked by viral immune evasion genes. Indeed, many viruses harbor genes which can inhibit or blunt the production of IFN-I/III by infected cells. Since IFN-I/III responses are critical for host resistance to viral infections both in mice and humans, it was therefore inferred that pDC have a critical beneficial role in antiviral defense by promoting the production of high levels of the cytokines in the face of the viral immune evasion mechanisms acting in infected cells. However, in mice, data demonstrating a non-redundant and critical role of DC for antiviral defense are scarce. Moreover, humans bearing primary immune deficiencies leading to the loss of pDC IFN-I/III production do not show enhanced susceptibility to viral infections. Thus, whether pDC play a unique and critical role promoting host defense during viral infections is still a matter of debate in mice and even more in humans. More generally, to the best of my knowledge, how pDC are activated as compared to other mononuclear phagocytes in vivo during an acute viral infection in humans has not been studied yet, in part due to the difficulty in diagnosing the disease and harvesting blood samples early enough after infection. The current paper takes advantage of unique and rare blood samples harvested in three patients during acute infection with the zika Virus (ZIKV) in order to address these questions. Specifically, the activation state of the pDC of these patients were characterized *ex vivo* by gene expression profiling, as compared to pDC from matched healthy controls as well as with other immune cell populations isolated from the same blood samples, including two other types of mononuclear phagocytes, monocytes and so-called myeloid dendritic cells (mDCs), as well as B lymphocytes, natural killer (NK) cells, CD4+ T cells and CD8+ T cells. A variety of computational analyses were then performed on the gene modules found to be differentially expressed between at least on cell type from the patients and its counterpart from healthy controls, to infer the associated biological processes, their molecular regulation and their potential relevance to the orchestration of antiviral immunity and to the defense of the host.

This paper represents a major “tour de force” by combining the study of very unique, precious and interesting clinical samples with cutting-edge technological approaches including micro-bulk RNA sequencing on 7 different immune cell populations and deep computational analyses of the data. This study will likely attract considerable attention in the broad community of researchers interested in antiviral immunity and in the biology of pDC.

The major claims of the paper are the following.

Claim #1) Contrary to other mononuclear phagocytes (monocytes and mDC), pDC are resistant to CHIKV infection, in part due to their downregulation of ZIKV dependency genes, and also due to their unique upregulation of a specific set of restriction factors.

Claim #2) Contrary to other mononuclear phagocytes (monocytes and mDC), pDC mount strong IFN-I responses to ZIKV infection, likely due to their ability to sense the virus through TLR7, without being infected and thus without being affected by the immune evasion genes expressed in infected mononuclear phagocytes.

Claim #3) Social network analysis of immune cells following ZIKV infection highlight a unique and central role of pDC in the orchestration of antiviral immunity.

A major limitation of the paper is that it stops at the stage of inferences from the analysis of the gene expression profiling, without formulating precise enough hypothesis and testing them experimentally. Hence, additional experiments are required to increase the significance/impact of the study by experimentally testing some of functions proposed to be unique for pDC and to promote antiviral defense.

Moreover, a number of conclusions in the paper are overstated, either due to inconsistencies in the use of the results of the statistical analyses performed to identify differentially expressed genes, or because of insufficient quantitation of the differences reported between the different cell types studied. Hence, additional analyses or experiments, and some rewriting, are required to strengthen or modulate some of the conclusions of the paper.

Finally, the main conclusion about the unique role of pDC should also be modulated, by discussing them in a broader context, considering that the authors have not studied another type of DC known to be critical for antiviral defense, namely type 1 cDC (cDC1). Indeed, cDC1 share with pDC their broad resistance to viral infections [PMID: 28783704], the ability to produce IFN-B1 and IFN-III without being infected (in response to TLR3 triggering) [PMID: 20975040], and have been clearly and indisputably demonstrated in mice to be absolutely critical to promote protective antiviral innate and adaptive immune responses [PMID: 19008445; 22505455; 22505458; 26296422; 26297566; 27692611; 28190711; 30409884]. Hence, a number of the characteristics of in vivo pDC responses to acute ZIKV infection in humans may not be unique to this cell type but largely shared by cDC1, with both overlapping and complementary roles of these two cell types in the global orchestration of protective antiviral immune responses.

We thank the reviewer for her/his extensive comments regarding our manuscript and are very pleased that s/he thinks our manuscript 'will likely attract considerable attention in the broad community of researchers interested in antiviral immunity and in the biology of pDCs'. We believe these comments have helped us substantiate our conclusions and strengthen our manuscript. Please see below for a detailed point-by-point response to the comments and concerns made:

MAJOR POINTS.

Major point #1) More precise hypotheses should be formulated from the computational analyses performed, and then tested experimentally.

1-1) The authors should test whether the knock-down of IFIH1, IFITM3 or TMEM2, or enforced ectopic expression of EMC7, AXL or IDO1, in pDC increases their susceptibility to ZIKV infection.

We thank the reviewer for highlighting this important point, and we agree with the reviewer's view. Despite the considerable technical difficulties of manipulating the gene expression of primary pDCs *in vitro*, we performed knock-down and overexpression experiments for 6 candidate genes to assess their role on ZIKV replication. Our results identified 3 interferon-stimulated candidate genes that, upon individual or combined siRNA-mediated gene silencing, induced substantial increases of ZIKV replication in pDCs. Interestingly, strongest effects were noted after combined gene silencing, suggesting a synergistic inhibitory effect on ZIKV replication. These data are now shown in the new figure 3f and the new supplemental figure 3.

We have also attempted to induce enforced overexpression of selected genes in primary pDCs, as suggested by the reviewer. However, under multiple different experimental conditions, ectopic enforced expression of these genes resulted in unacceptable toxicity that precluded functional evaluations.

1-2) The authors should examine how pDC depletion from healthy donor PBMCs before their exposure to ZIKV alters the transcriptional reprogramming of other immune cell types, in particular B cells, to experimentally test their hypothesis of a central role of pDCs in immune defense during ZIKV acute infection through their establishment of immunological circuits and other cells, in particular with B cells to promote their activation.

The reviewer makes an excellent point. In response to these comments, we have conducted pDC depletion experiments, followed by an evaluation of the susceptibility of other subsets to ZIKV infection and the immune response of B cells. We observed that ZIKV-induces activation of B cells, measured by CD69 expression, dependent at least partially on the presence of pDCs. Furthermore, depletion of pDC increases the susceptibility of the other subsets to ZIKV infection and lower the production of IFN-

α . Together, these results support our original observation that pDCs have a critical role for orchestrating the human response to ZIKV.

Major point #2) A number of conclusions in the paper are overstated due to insufficient rigor in data description and interpretation. A number of results that are not significant according to the figures or the data tables are discussed in the main text as if they are. This is not appropriate and must be corrected. Please find below a non-exhaustive list of examples.

We apologize for the lack of clarity in the previous submission and have revised the manuscript accordingly.

2-1) Line 204, contrary to what is written in the main text, the direction of the modulation of the “antigen presentation” canonical pathway in B cells and in mDCs is undetermined (Figure 2a) in the computational analysis, and thus not documented as inhibited.

We thank the reviewer for her/his comments and have changed the manuscript accordingly in line 204.

2-2) It is not clear how the authors have selected the terms used to describe the results of the functional annotations of the DEGs (Figure 2b), since “Type I IFN immune responses” or “effector immune cell recruitment” are not used in the functional annotation terms included in the corresponding figure and supplemental table. Please use the ontology directly provided from the database mined for the computational analysis. If an additional layer of interpretation can be added upon further mining of the gene lists, to refine the nature of the biological process at play, please justify precisely how this was achieved.

In response to the reviewer’s comments, we have adjusted the language to highlight the gene ontology terms provided by the Ingenuity pathway analysis (IPA).

2-3) Line 215-216, it is not true that the involvement in transcriptional regulation was markedly less obvious in alternative cell populations than in pDCs for all of the cytokine upstream regulators listed. For example, Figure 2c clearly shows as strong a significance for IFN β , IFN α 2, Interferon alpha in B cells and CD8+ T cells than in pDC, and this is also the case for IL15, IL3, EBI3, TNFSF10, IFNL1 and OSM, etc... The difference stand out mostly for the upstream regulators that are central to the heatmap, including CCL5, TNF and CSF1.

We have now adjusted the language of the manuscript to emphasize that cytokine upstream regulators related to TNF, CSF1, CSF3, CSF2, IL-4 and EDN1 were most profoundly altered between pDC and alternative immune cell populations. That said, we would like to point out that IL-1 β , IFN- γ , IL-5 also show marked differences between pDCs and alternative cell populations. The z-scores for these cytokines are listed in the table below.

Upstream regulators	pDC	mDC	monocytes	CD8	CD4	B cells
IL1B	5,50	-0,47	-0,98	3,00	N/A	0,46
IFNG	4,73	-3,73	-1,47	2,28	-0,79	0,82
TNF	3,92	-1,56	-2,40	2,48	-1,30	-0,45
IL5	4,30	1,83	-2,71	0,82	-0,56	2,89

2-4) Lines 235-236. When taking into account the statistical analysis, it is not true that EMC7, AXL and IDO1 were downregulated in pDC but upregulated in the other cell populations. The changes in expression of EMC7 and AXL in pDC are not significant. The only significant changes are: an up-regulation of AXL and IDO1 in mDC, and an up-regulation of EMC7 in B cells and monocytes. Of note, the data for IDO1 in pDC in the supplemental table is missing, which must be corrected. Rather, when looking at figure 3b, it appears that one of the characteristic of pDC response in acute ZIKV infection is their lack of significant upregulation of any of the known ZIKV dependency genes, whereas a number of these genes are upregulated in B cells, mDC and monocytes. A Gene Set Enrichment Analysis (GSEA) would be welcome here to compare each immune cell population between patients and controls, as well as to compare the pDC to each of the other cell populations in the patients, for enrichment in the gene set corresponding to the list of the known ZIKV dependency genes illustrated as Figure 3b. This would allow to better quantitate the global expression changes of the ZIKV dependency genes in patients as compared to controls, and the relative expression of this gene module between cell types in patients.

The manuscript has been changed to describe changes in expression of EMC7, AXL and IDO1 in more detail. We have now included a GSEA analysis, as requested by the reviewer. Consistent with our original hypothesis, our results show de-enrichment of pDCs with viral dependency genes. We would like to thank the reviewer for this very helpful comment!

2-5) Lines 251-252. When taking into account the statistical analysis, it is not true that all the ZIKV restriction factors listed are upregulated in pDCs in contrast to alternative cell compartments. IFIH1, IFITM3 and TMEM2 were indeed significantly upregulated in pDCs and not significantly changed in B cells, monocytes and mDCs. However, IFI27 was significantly upregulated in mDCs and IFIT3 in monocytes but the expression of these genes did not change significantly in pDCs. The expression of TNFAIP2, TNFAIP3, IRF1 and OAS3 did not change significantly in pDCs. Here again, a GSEA would be welcome to better quantitate the global expression changes of the ZIKV restriction factor gene module in patients as compared to controls, and the relative expression of this gene module between cell types in patients.

We have now changed the manuscript to describe transcripts which showed significantly different expression between pDCs and alternative cell populations, and other transcripts for which only a trend is visible. In addition, we have conducted a new GSEA to determine to what extent differentially-expressed

genes for each immune cell subsets are enriched for Interferon-stimulated genes. This analysis showed a significant enrichment of pDCs with ISG.

2-6) Lines 253-255, the sentence "Moreover, a computational analysis of upstream regulators governing signatures of DEGs supported a critical role of type I IFN responses in pDCs from ZIKV-infected patients, in contrast to mDCs in which type I IFN signatures appeared to be reduced (Figure 3f)" needs to be supported by a more precise, quantitative, analysis of the data.

Unfortunately, the network analysis originally provided in Figure 3f does not allow for more quantitative data analysis. In response to the reviewer's comments, we have decided to de-emphasize this dataset and move it to the paper supplements (new Supplemental Figure 1a).

2-7) Lines 276-277, the sentence "no such activation of IFN-dependent immune pathways was seen in gene expression signatures from *in vitro* infected and patient-derived mDCs (Fig. 4c)." is an overstatement considering the positive z-score for "Interferon signaling" in mDC cells at 48 hours after *in vitro* stimulation, and *ex vivo*. Here again, a GSEA would be welcome to better quantitate the global expression changes of the ISG module in patients as compared to controls, and the relative expression of this gene module between cell types in patients.

We have modulated the language regarding the IFN-dependent immune pathways in *in vitro* infected and patient-derived mDCs. As suggested by the reviewer, we have conducted a GSEA to analyze enrichment of *in vitro* infected cells with viral dependency genes or ISG. Given the relatively small numbers of DEGs in *in vitro* infected cells, this GSEA did not allow to identify significant enrichment or de-enrichment. We therefore did not include these results into the manuscript.

Minor points.

Minor point #1) The main conclusion about the unique role of pDC should be modulated. First, on many aspects, B cells appear to behave to a large extent similarly to pDC. Second, the authors have not examined the behavior of cDC1 which may well share even more characteristics with pDC in their response to ZIKV acute infection. Hence, without further studies, the authors cannot state that their "results indicate a remarkably unique role of pDCs in the human immune response to ZIKV" as written in their title and discussion.

We agree with the reviewer's comment and changed the title and the discussion – we now argue that pDCs have a distinct role for host immune responses, which we believe is well supported by the data of this paper. We propose to conduct an evaluation of cDC1 responses to ZIKV infection in future studies.

Reviewer #2 (Remarks to the Author):

In this manuscript, the authors utilize a systems biology-based approach to understand the biological networks within immune cells isolated from ZIKV-infected individuals. As compared to other myeloid cells, plasmacytoid DCs (pDCs) displayed innate immune activation, characterized by the induction of type I interferon (IFN). While this study is interesting, the findings are largely descriptive with little to no biological validation. The study was difficult to read and the figures were largely a collection of heatmaps and “hairballs” of gene networks. In addition, there were a number of concerns with this study:

1. It is unclear if equal cell numbers were used across the transcriptional profiling studies. This is especially important with cells, such as s B cells.

We have added the following statement to the manuscript to address this.

“We have adjusted the amount of RNA input to 10ng for all subsets prior processing to RNAseq.”

2. This analysis is based on a single infection timepoint and this information is not readily found within the text of the results section. Rather than including a reference, this pertinent information should be included in this manuscript.

We thank the reviewer for raising this issue. The patient information was reported in our previous study (PMID:29262327). As suggested, we have addressed this issue by adding a supplemental table S9 to the manuscript.

3. The canonical pathway analysis is rather confusing and misleading. Many of these pathways/biological functions can differ by only 1 or 2 genes. This is not reflected in the data presented in Figure 2. Moreover, precisely how are functional pathways predicted to be up or down regulated simply based on mRNA expression patterns? The authors need to provide biological validation of these data rather than predictive models.

We used the Ingenuity Pathway Analysis (IPA), a well-established and frequently-used computational algorithm to analyze our transcriptional data which can predict the direction of the functional pathways based on mRNA expression. We have now added an additional citation to refer readers to the complex algorithms underlying IPA. We agreed that biological validation is needed. In Figure 3F and in Figure 6, we have included functional validation studies to address this.

4. It is not clear what the authors mean by “distinct role of pDCs” to ZIKV infection. pDCs are inherently a distinct cell population and are known to be major inducers of type I IFN. Moreover, it is also well known that flaviviruses do not infect pDCs.

Our new data show that despite their reduced susceptibility to ZIKV infection, pDCs have the capacity to protect other cells from ZIKV infection to some extent, likely through type I interferon expression. Moreover, our computational analysis suggested an interaction between pDCs and B cells; this observation has been confirmed in *in vitro* model with primary pDCs and B cells, as shown in Figure 6.

5. The authors need to provide an in-depth analysis of ZIKV infection of mDCs and pDCs. Moreover, ZIKV is not a poly-adenylated virus. It is unclear how the authors were able to capture ZIKV RNA using an oligo-DT primed cDNA reaction for mRNA-seq. These results are not reliable and do not reflect ZIKV replication. What is the purity of the pDC population? Are pDCs infected in ZIKV-infected patients?

We thank the reviewer for bringing up these issues and for giving us an opportunity to explain our approach in detail. Our ability to detect ZIKV RNA with an oligo-DT primer is related to the presence of an Adenine-rich region in the ZIKV NS5 gene that is susceptible to annealing with the oligo-DT primer. Consistent with this observation, most of the ZIKV RNA-Seq reads mapped to the NS5 region.

We would like to point out that a similar analysis of ZIKV RNA with oligo-DT primers has previously been performed by other groups (e. g. *RNA-Seq analysis of chikungunya virus infection and identification of granzyme A as a major promoter of arthritic inflammation. PLOS Pathogens 2017*; *N6-Methyladenosine in Flaviviridae Viral RNA Genomes Regulates Infection. Cell Host & Microbe 2016*; *ZIKV infection regulates inflammasomes pathway for replication in monocytes. Scientific Reports, 2017*). These references are now added to our manuscript to support our analysis.

Cell sorting of immune cells subsets for RNAseq analysis was performed using a BD FACS ARIA with a purity of >95%.

With regards to the infection of pDC in ZIKV-infected patients, we have previously demonstrated that we were only able to detect ZIKV mRNA in mDCs, not in pDC following in vitro exposure of cells to ZIKV (see Figure 4a). The same was true when analyzing pDC and mDC from individuals with ZIKV infection (see *Transcriptional Changes during Naturally Acquired Zika Virus Infection Render Dendritic Cells Highly Conducive to Viral Replication. Cell Reports 2017*).

6. The findings that JAK/STAT pathways are critical hubs for ZIKV infection are not a new concept. Many flaviviruses are highly sensitive to type I IFN and it is no surprise that replication/infection corresponds to IFN-related signatures.

We agree with the reviewer's comments about the critical role of JAK/STAT pathways and type I IFN are not a new concept. However, multiple prior studies suggested that ZIKV can actively inhibit type I IFN responses to reprogram host immunity in favor of the virus. Ours is the first study showing directly in patient-derived cells that human pDCs display a higher activation and inflammatory response during acute ZIKV infection. Moreover, we have demonstrated in figure 6C that pDCs can resist ZIKV infection and activate B cells through type I Interferon production and signaling.

7. Much of this manuscript is based on predictive computational modeling with little to no biological validation studies. It is not surprising that immune cells interact with each other. Several groups have shown the importance of pDCs in bridging innate and adaptive immune functions.

We have now added an *in vitro* validation of genes involved in the protection of pDCs to ZIKV replication in supplemental figure S3 and have added functional assays to investigate how pDCs and B cells interact to induce B cell activation (Figure 6).

Reviewers' comments:

Reviewer #1 (Remarks to the Author):

The authors have made obvious efforts to attempt to address most of my concerns and suggestions, not only by editing the main text of the paper but also by performing new experiments, including candidate gene knock-down in primary pDCs. These modifications have further improved the paper, including by adding experimental data supporting the hypotheses inferred from the analysis of the gene expression profiling of the different cell types examined upon isolation from patients infected with the Zika virus versus matched controls, or upon in vitro exposure to the Zika virus of cells isolated from healthy blood donors. A few minor issues need to be addressed as listed below.

Minor issues.

Page 3, line 85, a sentence is incorrect and needs correction. Indeed, whereas it is written "For instance, in myeloid dendritic cells (mDCs), ZIKV can inhibit phosphorylation and promote proteasomal degradation of human STAT2, a key downstream regulator of IFN-I responses 19", reference 19 did not study ZIKV infection of dendritic cells but used 293 T cells, Verol cells and Fibroblasts. Hence, either another reference demonstrating the same effect in mDCs must be added, or the sentence must be modified by replacing "in myeloid dendritic cells (mDCs)" by "in target cells".

Page 8, line 221: the title "ZIKV qRT-PCR and qPCR" might be better rewritten as "qPCR for selected host genes and ZIKV qRT-PCR".

Page 9, lines 269-271, the sentence "Consistent with these results, functional annotations of the DEGs in B cells, CD4 T cells, monocytes, and mDCs were predicted to be inhibited or negatively enriched (Fig. 2b)" might be better rewritten as "Consistent with these results, no functions were predicted to be induced in B cells, CD4 T cells, monocytes and mDCs based on annotation enrichment analysis of their DEGs using IPA (Fig. 2b)."

Page 9, line 273 it is written "secretion of innate cytokines (IL-8)" whereas the title of IPA pathway is "IL-8 signaling". The authors should thus justify their interpretation of the enrichment analysis by listing the specific DEGs responsible for this enrichment analysis, which should include the gene encoding IL-8 itself. If this is not the case, and the DEGs responsible for this enrichment are rather genes induced by IL-8, then the authors need to go back to using the exact IPA nomenclature, i.e. "IL-8 signaling".

All over the paper, for GSEA analyses, likewise the authors are documenting the NES and not the ES, it is the FDR q-value that must be used to document significance, and not the nominal p-value.

Page 10, lines 302-303, a sentence is incorrect and must be edited. The authors write "in contrast to B cells, mDCs and monocytes in which no significant enrichment or de-enrichment of ZDGs was observed (Supplementary Table 10)", whereas B cells rather behave a bit like pDCs, since ZDGs clearly tend to be decreased in these cells (NES=-1.350834, and FDR= 0.05753564 which is very close to the significance threshold of 0.5).

Page 11, line 326, "OA3" must be replaced by "OAS3" and I suppose that "TNFAIP2" must be replaced by "TNFAIP3".

Page 11, lines 330-331, the author state "a trend for negative enrichment with ISG was also observed for the monocyte compartment (NES of -0.88, p=0.65)" despite the fact that the nominal p-value is not at all close to significance and the FDR q-value is similar (0.6314200). This interpretation is paradoxical with the statement above that no significant enrichment or de-enrichment of ZDGs was observed in B cells whereas both the NES, p-value and FDR q-value were actually indicative of a much more convincing trend of de-enrichment in that case.

Page 14, line 432-433, in the sentence "the early activation marker CD69 in response to ZIKV infection of total PBMC; however, this upregulation was greatly diminished after experimental depletion of pDCs", please replace "greatly" by "significantly". Indeed, while the decrease is significant (but only with $0.01 < p < 0.05$), it should not be qualified as "great" considering the very large overlap of the values across all experimental conditions, and the fact that the percentage of CD69 induction observed when co-culturing PBMC, pDC and ZIKV as compared to PBMC and pDC or PBMC alone is only moderately increased, and this induction is decreased by only ~half in the absence of pDC.

Figure 1c, for NK cells, it does not make sense to show a heatmap of 100 genes since only 55 genes are significantly deregulated. The heatmap of DEGs for NK cells should focus only on the DEGs complying with the selection criteria used for the authors as illustrated on panel 1, Figure B, i.e. 46 up-regulated DEGs and 9 down-regulated DEGs.

Reviewer #2 (Remarks to the Author):

This is a revised manuscript that describes the transcriptional landscape of innate and adaptive immune cells within three individuals that were naturally infected with ZIKV. As expected, pDCs displayed the most dramatic type I IFN-related gene signature that correlated with the downregulation of genes that support ZIKV infection. Using computational modeling, the authors find that pDCs may be interacting with B cells, however, the importance of this interaction is not made clear in this study. In general, the authors have responded to the previous reviewer's comments through changes in the text and including additional experimental data. However, there are still several outstanding concerns that the authors need to address:

Major concerns:

1. A previous comment was made regarding normalization based on cell numbers. The authors did not provide clarity on this issue but rather mentioned that the RNA was adjusted to 10 ng for all subsets. How many cells (individual cell subsets) does 10 ng RNA represent?
2. Are pDCs a target cell of ZIKV infection in vivo (not in vitro infections)? What is ZIKV mRNA? ZIKV does not encode mRNA but rather is a plus-strand virus that can be translated.
3. ZIKV is not a polyadenylated virus, this using oligoDT to capture viral RNA is an inaccurate method. While other studies have utilized this method, this is an unequivocally incorrect method. The authors need to demonstrate that capturing ZIKV RNA through this method is as sensitive as capturing using virus-specific primers.
4. How do pDCs trigger type I IFN in response to ZIKV? The authors did not include the appropriate UV-inactivated controls to ensure that no contaminants within the viral preparation are responsible for aberrantly triggering pDCs.
5. What is the proportion of pDCs and mDCs that are alive in culture after 24-48 hours?
6. Figure 4: It is unclear if the authors compared against time-matched uninfected control samples to determine changes in gene expression.
7. Figure 6: The authors need to demonstrate virus replication (or the lack thereof) through infectious release assays (plaque assays or focus-forming assays). Alternatively, the authors should determine whether they can detect minus-strand. It is unclear from this study whether the authors are measuring input virus or replicating virus.

8. Figure 6: Negative controls are not included in this assay. Namely, UV-inactivated virus controls.

9. What do the authors mean by the percentage of CD86? Histograms should be shown.

10. While the authors claim that they show pDCs interacting with B cells in Figure 6, this data is rather confusing. Are the cells physically interacting? What is the negative control cell in this experiment (one that was not enriched in the computational models)? Which cells are producing type I IFN?

Reviewer #1

The authors have made obvious efforts to attempt to address most of my concerns and suggestions, not only by editing the main text of the paper but also by performing new experiments, including candidate gene knock-down in primary pDCs. These modifications have further improved the paper, including by adding experimental data supporting the hypotheses inferred from the analysis of the gene expression profiling of the different cell types examined upon isolation from patients infected with the Zika virus versus matched controls, or upon in vitro exposure to the Zika virus of cells isolated from healthy blood donors. A few minor issues need to be addressed as listed below.

We very much appreciate the dedicated and thoughtful comments from Reviewer 1 throughout the entire review process – these suggestions have helped us to considerably improve this manuscript.

Minor issues.

Page 3, line 85, a sentence is incorrect and needs correction. Indeed, whereas it is written “For instance, in myeloid dendritic cells (mDCs), ZIKV can inhibit phosphorylation and promote proteasomal degradation of human STAT2, a key downstream regulator of IFN-I responses 19”, reference 19 did not study ZIKV infection of dendritic cells but used 293 T cells, Verol cells and Fibroblasts. Hence, either another reference demonstrating the same effect in mDCs must be added, or the sentence must be modified by replacing “in myeloid dendritic cells (mDCs)” by “in target cells”.

Thanks for picking this up – this has been corrected.

Page 8, line 221: the title “ZIKV qRT-PCR and qPCR” might be better rewritten as “qPCR for selected host genes and ZIKV qRT-PCR”.

Thanks for this suggestion, which we have incorporated into the manuscript.

Page 9, lines 269-271, the sentence “Consistent with these results, functional annotations of the DEGs in B cells, CD4 T cells, monocytes, and mDCs were predicted to be inhibited or negatively enriched (Fig. 2b)” might be better rewritten as “Consistent with these results, no functions were predicted to be induced in B cells, CD4 T cells, monocytes and mDCs based on annotation enrichment analysis of their DEGs using IPA (Fig. 2b).”

This is an excellent suggestion- many thanks.

Page 9, line 273 it is written “secretion of innate cytokines (IL-8)” whereas the title of IPA pathway is “IL-8 signaling”. The authors should thus justify their interpretation of the enrichment analysis by listing the specific DEGs responsible for this enrichment analysis, which should include the gene encoding IL-8 itself. If this is not the case, and the DEGs responsible for this enrichment are rather genes induced by IL-8, then the authors need to go back to using the exact IPA nomenclature, i.e. “IL-8 signaling”.

IL-8 is a DEGs for B cells, CD8 T cells, mDCs and pDCs

We would like to thank the reviewer for this suggestion, IL-8 is a DEGs in pDCs and IL-8 signaling is predicted to be upregulated. We have incorporated this into the manuscript.

All over the paper, for GSEA analyses, likewise the authors are documenting the NES and not the ES, it is the FDR q-value that must be used to document significance, and not the nominal p-value.

This has been corrected as suggested.

Page 10, lines 302-303, a sentence is incorrect and must be edited. The authors write “in contrast to B cells, mDCs and monocytes in which no significant enrichment or de-enrichment of ZDGs was observed (Supplementary Table 10)”, whereas B cells rather behave a bit like pDCs, since ZDGs clearly tend to be decreased in these cells (NES=-1.350834, and FDR= 0.05753564 which is very close to the significance threshold of 0.5).

This has been corrected as suggested.

Page 11, line 326, “OA3” must be replaced by “OAS3” and I suppose that “TNFAIP2” must be replaced by “TNFAIP3”.

These issues have also been corrected.

Page 11, lines 330-331, the author state “a trend for negative enrichment with ISG was also observed for the monocyte compartment (NES of -0.88, p=0.65)” despite the fact that the nominal p-value is not at all close to significance and the FDR q-value is similar (0.6314200). This interpretation is paradoxical with the statement above that no significant enrichment or de-enrichment of ZDGs was observed in B cells whereas both the NES, p-value and FDR q-value were actually indicative of a much more convincing trend of de-enrichment in that case.

This has been corrected as suggested.

Page 14, line 432-433, in the sentence “the early activation marker CD69 in response to ZIKV infection of total PBMC; however, this upregulation was greatly diminished after experimental depletion of pDCs”, please replace “greatly” by “significantly”. Indeed, while the decrease is significant (but only with $0.01 < p < 0.05$), it should not be qualified as “great” considering the very large overlap of the values across all experimental conditions, and the fact that the percentage of CD69 induction observed when co-culturing PBMC, pDC and ZIKV as compared to PBMC and pDC or PBMC alone is only moderately increased, and this induction is decreased by only ~half in the absence of pDC.

We have changed the manuscript as suggested.

Figure 1c, for NK cells, it does not make sense to show a heatmap of 100 genes since only 55 genes are significantly deregulated. The heatmap of DEGs for NK cells should focus only on the DEGs complying with the selection criteria used for the authors as illustrated on panel 1, Figure B, i.e. 46 up-regulated DEGs and 9 down-regulated DEGs.

Figure 1c has been revised accordingly.

Reviewer #2:

This is a revised manuscript that describes the transcriptional landscape of innate and adaptive immune cells within three individuals that were naturally infected with ZIKV. As expected, pDCs displayed the most dramatic type I IFN-related gene signature that correlated with the downregulation of genes that support ZIKV infection. Using computational modeling, the authors find that pDCs may be interacting with B cells, however, the importance of this interaction is not made clear in this study. In general, the authors have responded to the previous reviewer's comments through changes in the text and including additional experimental data. However, there are still several outstanding concerns that the authors need to address:

Major concerns:

1. A previous comment was made regarding normalization based on cell numbers. The authors did not provide clarity on this issue but rather mentioned that the RNA was adjusted to 10 ng for all subsets. How many cells (individual cell subsets) does 10 ng RNA represent?

Cell numbers have now been provided in a new supplemental Table 11.

2. Are pDCs a target cell of ZIKV infection in vivo (not in vitro infections)? What is ZIKV mRNA? ZIKV does not encode mRNA but rather is a plus-strand virus that can be translated.

ZIKV RNA was measured in pDC from three study subjects and were reported in our previous manuscript in which the same study persons were analyzed. In contrast to mDC, no ZIKV RNA was detected in pDC. The manuscript has been edited to reflect this, and the previous publication is cited.

3. ZIKV is not a polyadenylated virus, this using oligoDT to capture viral RNA is an inaccurate method. While other studies have utilized this method, this is an unequivocally incorrect method. The authors need to demonstrate that capturing ZIKV RNA through this method is as sensitive as capturing using virus-specific primers.

We have used virus-specific primers to analyze ZIKV RNA in in vitro infected cells; the data are now incorporated in the updated manuscript (Figure S2A).

4. How do pDCs trigger type I IFN in response to ZIKV? The authors did not include the appropriate UV-inactivated controls to ensure that no contaminants within the viral preparation are responsible for aberrantly triggering pDCs.

IFN secretion and surface expression of the activation marker CD69 and CD86 on B cells co-cultured with pDC have now been evaluated in cells exposed to UV-inactivated ZIKV. The results are now shown in the revised Figure S5. In line with previous results in the context of other flaviviruses¹⁻⁴, we observed that UV-treated ZIKV did not fully abrogate cell-intrinsic immune responses, most likely because fragmented ZIKV RNA segments can still be recognized by innate immune sensors.

5. What is the proportion of pDCs and mDCs that are alive in culture after 24-48 hours?

Frequencies of live cells after 24 hours of culture are now mentioned in the manuscript.

6. Figure 4: It is unclear if the authors compared against time-matched uninfected control samples to determine changes in gene expression.

Time-matched comparisons were used to evaluate changes in gene expression between infected and uninfected cells; the manuscript has been edited to reflect this.

7. Figure 6: The authors need to demonstrate virus replication (or the lack thereof) through infectious release assays (plaque assays or focus-forming assays). Alternatively, the authors should determine whether they can detect minus-strand. It is unclear from this study whether the authors are measuring input virus or replicating virus.

Minus-strand ZIKV RNA has now been measured; results are shown in Figure S5B. We would also like to refer the reviewer to our previous study⁵ (PMID: 29262327, ref 16 in current manuscript) showing the presence of minus-strand ZIKV RNA, indicating active replication of ZIKV RNA in PBMC samples.

8. Figure 6: Negative controls are not included in this assay. Namely, UV-inactivated virus controls.

IFN secretion and surface expression of the activation marker CD69 and CD86 on co-cultured B cells have now been evaluated in cells exposed to UV-inactivated ZIKV. The results are now shown in the revised Figure S5. Please see also our comments on issue 4.

9. What do the authors mean by the percentage of CD86? Histograms should be shown.

Percentages were meant to reflect "Proportions of positive cells" – this has now been corrected. Histograms are now shown for visual enhancement in Figure S5.

10. While the authors claim that they show pDCs interacting with B cells in Figure 6, this data is rather confusing. Are the cells physically interacting? What is the negative control cell in this experiment (one that was not enriched in the computational models)? Which cells are producing type I IFN?

During co-culture of B cells with ZIKV-infected pDC, we noted upregulation of CD69 on B cells, confirming a functional interaction between B cells and pDC that we predicted based on computational modeling. In contrast, there was no increase of CD69 expression on T cells and NK cells during co-culture with ZIKV-infected pDC, consistent with our computational modeling data which failed to highlight functional connections between pDC and T/NK cells. These data are now summarized in Figure 6d.

In order to further confirm this interaction between B and pDCs, we performed co-culture experiments between pDC and B cells in the presence of IFN- α -blocking antibodies; we observed that these antibodies effectively inhibited activation of B cells during co-culture with ZIKV-infected pDC (Figure 6e). In addition, we noted that separation of pDC and B in transwell co-culture systems also abrogated activation of B cells in response to ZIKV-infected pDC. We therefore propose that the observed interactions between pDC and B cells rely both on soluble type I IFN and physical cell contacts, and IFN secretion by pDC most effectively activates B cells

located in immediate anatomical proximity to pDC. The manuscript has been edited to express this view.

References

1. Rodriguez-Madoz, J. R. *et al.* Inhibition of the Type I Interferon Response in Human Dendritic Cells by Dengue Virus Infection Requires a Catalytically Active NS2B3 Complex. *Journal of Virology* **84**, 9760–9774 (2010).
2. Wang, J. P. *et al.* Flavivirus Activation of Plasmacytoid Dendritic Cells Delineates Key Elements of TLR7 Signaling beyond Endosomal Recognition. *The Journal of Immunology* **177**, 7114–7121 (2006).
3. Aguirre, S. *et al.* DENV Inhibits Type I IFN Production in Infected Cells by Cleaving Human STING. *PLoS Pathog* **8**, (2012).
4. Silva, M. R. *et al.* Suppression of chikungunya virus replication and differential innate responses of human peripheral blood mononuclear cells during co-infection with dengue virus. *PLOS Neglected Tropical Diseases* **11**, e0005712 (2017).
5. Sun, X. *et al.* Transcriptional Changes during Naturally Acquired Zika Virus Infection Render Dendritic Cells Highly Conducive to Viral Replication. *Cell Rep* **21**, 3471–3482 (2017).

Reviewers' comments:

Reviewer #2 (Remarks to the Author):

This is a revised manuscript in which the authors have provided additional data and revisions to the text that were raised by the previous reviewers. There are still major concerns with this manuscript that need to be addressed:

1. The cell numbers for the RNAseq assay need to be incorporated into the main figure. This is a critical piece of information that is central to interpreting the data presented in Figure 1.
2. Where are the flow cytometry plots showing the gating strategy used for isolating these individual cell populations presented in Figure 1?
3. What methodology was used to detect ZIKV negative-strand RNA? There are no methods described in this paper to demonstrate how the authors specifically detected negative-strand vs positive-strand RNA.
4. Figure 3: It is unclear what the authors mean by "we observed that UV-treated ZIKV did not fully abrogate cell-intrinsic immune responses, most likely because fragmented ZIKV RNA segments can still be recognized by innate immune sensors" What are fragmented ZIKV RNA segments? ZIKV is a single-stranded RNA virus. The authors need to demonstrate HOW ZIKV is triggering an immune response in pDCs.
5. Figure S5: It is unclear why this critical information is included as supplemental material rather than included in the main figure panels. The data as presented in Figure 3 is uninterpretable without showing the UV-inactivated controls in the same panel.
6. Figure 4D: The authors state that "These studies further identify components of IFN signaling pathways, such as JAK/STAT and MAPK kinases, as critical hubs for the transcriptional response to ZIKV infection in pDCs." At this point, these findings are strictly computational and there is no experimental data to demonstrate/support these findings.
7. Figure 4A: The authors did not address a critical question raised by the previous reviewer regarding the detection of viral RNA sequences by RNA-seq assay. ZIKV is not a polyadenylated virus. These reads are likely a result of mispriming and are not reliable data. These data should be removed from the analysis.

Reviewer #2

This is a revised manuscript in which the authors have provided additional data and revisions to the text that were raised by the previous reviewers. There are still major concerns with this manuscript that need to be addressed:

1. The cell numbers for the RNAseq assay need to be incorporated into the main figure. This is a critical piece of information that is central to interpreting the data presented in Figure 1.

These data have now been moved into Figure 1c.

2. Where are the flow cytometry plots showing the gating strategy used for isolating these individual cell populations presented in Figure 1?

These data have now been included in a new supplemental Figure 1a.

3. What methodology was used to detect ZIKV negative-strand RNA? There are no methods described in this paper to demonstrate how the authors specifically detected negative-strand vs positive-strand RNA.

The methods section has now been amended to include a description of how negative-strand ZIKV RNA was detected. This method is well established and was reported in prior manuscripts.

4. Figure 3: It is unclear what the authors mean by "we observed that UV-treated ZIKV did not fully abrogate cell-intrinsic immune responses, most likely because fragmented ZIKV RNA segments can still be recognized by innate immune sensors" What are fragmented ZIKV RNA segments? ZIKV is a single-stranded RNA virus. The authors need to demonstrate HOW ZIKV is triggering an immune response in pDCs.

The mechanism how ZIKV can be recognized by innate immune sensors has been extensively studied and described in prior work. As we have explained in our previous rebuttal letter, fragmented ZIKV RNA can occur as a result of UV treatment of viral particles. While our data show significant reduction in IFN secretion after exposure to UV-treated ZIKV (relative to untreated ZIKV), UV treatment did not fully abrogate IFN secretion. In our manuscript, we state:

"Of note, inactivation of ZIKV by UV light markedly reduced IFN- α mRNA expression in ZIKV-exposed pDCs, indicating that the observed effects were unrelated to non-specific contaminants in viral stocks"

This is from our perspective the most precise and accurate description of our findings.

5. Figure S5: It is unclear why this critical information is included as supplemental material rather than included in the main figure panels. The data as presented in Figure 3 is uninterpretable without showing the UV-inactivated controls in the same panel.

It is not clear to us why data in Figure S5 (data on cell-intrinsic recognition of ZIKV RNA, including UV controls) would be necessary as controls for Figure 3, which shows RNA-Seq data of in vivo infected cells (cell intrinsic immune responses in pDC in response to ZIKV infection, for which data with UV-inactivated ZIKV serves as a control, are shown in Figure 6). We believe the current organization of figures (with UV controls shown in supplemental Figure 5) is the best solution.

6. Figure 4D: The authors state that "These studies further identify components of IFN signaling pathways, such as JAK/STAT and MAPK kinases, as critical hubs for the transcriptional response to ZIKV infection in pDCs." At this point, these findings are strictly computational and there is no experimental data to demonstrate/support these findings.

We agree and have adjusted the manuscript to reflect the computational nature of these findings.

7. Figure 4A: The authors did not address a critical question raised by the previous reviewer regarding the detection of viral RNA sequences by RNA-seq assay. ZIKV is not a polyadenylated virus. These reads are likely a result of mispriming and are not reliable data. These data should be removed from the analysis.

While we disagree with this reviewer regarding the viral RNA sequences detected by RNA-Seq, we have removed the respective data to address this.